# Local and global chromatin interactions are altered by large genomic deletions associated with human brain development

Xianglong Zhang [1,2], Ying Zhang[3,5], Xiaowei Zhu[1,2], Carolin Purmann[1,2], Michael S. Haney [2], Thomas Ward[1,2], Arineh Khechaduri[1,2,6], Jie Yao [4,7], Sherman M. Weissman[3] & Alexander E. Urban[1,2]

Large copy number variants (CNVs) in the human genome are strongly associated with common neurodevelopmental, neuropsychiatric disorders such as schizophrenia and autism. Here we report on the epigenomic effects of the prominent large deletion CNVs on chromosome 22q11.2 and on chromosome 1q21.1. We use Hi-C analysis of long-range chromosome interactions, including haplotype-specific Hi-C analysis, ChIP-Seq analysis of regulatory histone marks, and RNA-Seq analysis of gene expression patterns. We observe changes on all the levels of analysis, within the deletion boundaries, in the deletion flanking regions, along chromosome 22q, and genome wide. We detect gene expression changes as well as pronounced and multilayered effects on chromatin states, chromosome folding and on the topological domains of the chromatin, that emanate from the large CNV locus. These findings suggest basic principles of how such large genomic deletions can alter nuclear organization and affect genomic molecular activity.

[1] Department of Psychiatry and Behavioral Sciences, Stanford University School of Medicine, Stanford 94304 CA, USA. [2] Department of Genetics, Stanford University School of Medicine, Stanford 94304 CA, USA. [3] Department of Genetics, Yale University, New Haven 06520 CT, USA. [4] Department of Cell Biology, Yale University School of Medicine, New Haven 06520 CT, USA. [5]Present Address: Department of Genetics and Genomics Sciences, Icahn School of Medicine at Mount Sinai & Sema4 NYC Laboratory, New York 10029 NY, USA. [6]Present Address: Vaccine and Infectious Disease Division, Fred Hutchinson Cancer Research Center, Seattle 98109 WA, USA. [7]Present Address: Sun Yat-sen University, Guangzhou 510080 Guangdong, China. These authors contributed equally: Xianglong Zhang, Ying Zhang, Xiaowei Zhu. Correspondence and requests for materials should be addressed to A.E.U. (email: aeurban@stanford.edu)

Two of the most exciting discoveries in human genetics of the past decade are that small-to-medium-sized copy number variants (CNVs) are very common in the human genome and that there is a group of large CNVs that are strongly associated with brain development and neuropsychiatric disorders, such as schizophrenia and the autism spectrum disorders (ASDs)[1,2]. These large CNVs are widely considered to be enticing points of entry to the analysis of the strong but complex genetic, molecular, and possibly even cellular, basis of these common disorders.

Large CNVs, typically sized from hundreds of thousands to millions of base pairs of genomic DNA sequence, were previously known to be in strong association with often severe but rare congenital malformations, or found in cancer genomes. It was a striking discovery when a series of studies[1,2] showed that there is a group of more than ten large CNVs that are strongly associated with aberrant brain development and a resulting neuropsychiatric phenotype such as schizophrenia or ASD. These large neuropsychiatric CNVs each encompass multiple genes and their effects across the various molecular levels of gene activity and regulation, and the connections from there to the clinical phenotypes, are complex and only poorly understood. For example, 22q11 deletion syndrome (22q11DS) is a disorder caused in the vast majority of cases by a heterozygous deletion of about 3 million base pairs spanning about 60 known genes on chromosome 22q11.2. It occurs in 1 per 3000–6000 live births[3]. The common phenotypes of 22q11DS include a large spectrum of congenital anomalies, for example of the facial structures and the immune and cardiovascular systems— and notably there is a strong association with several neurodevelopmental psychiatric disorders, in particular schizophrenia and ASD[2,4–7].

On the molecular level, these large neuropsychiatric CNVs have been mostly studied by focusing on the effects of individual genes or small groups of genes from within the CNV boundaries. Many very interesting insights have been gained using this approach.

However, these findings about individual genes fall short of explaining the full effects of the large CNVs. There already have been a number of transcriptome-wide studies that at least hint at certain network effects emanating from the large CNVs[8–12]. Which mechanisms mediate such transcription network effects is then the question. Furthermore, there are an increasing number of studies that show a potentially very important role of chromatin regulation in the molecular etiology of neuropsychiatric disorder[13–19].

Against this backdrop, we reasoned that it was worthwhile testing whether large CNVs with association with brain development might cause a disruption or at least alteration of one or several aspects of chromatin conformation, such as the distribution of regulatory chromatin marks, the long-range direct physical interactions between distant regions on one chromosome or between different chromosomes, or the higher-order chromatin domain structures that are defined by such marks or interactions. Such effects on these important layers of molecular regulation of gene activity would then constitute a basic principle by which large CNVs could transmit their presence to the machinery of cellular physiology.

Here we show, in a cohort of lymphoblastoid cell lines (LCLs) derived from patients with 22q11DS, that chromatin marks, chromatin domains, and long-range chromosome interactions are affected in several distinct ways by the large, common, and strongly disease-associated CNV on chromosome 22q11.2. We use the large CNV on 22q11.2 as a model to determine the generalizable principles along which large CNVs of this category can lead to changes to the various ways in which chromatin is ordered, using unbiased, genome-wide, sequencing-based assays for discovery. We then go on to show in a smaller number of LCLs from different patients that at least some of the same

observations can also be made for another neuropsychiatric large CNV on chromosome 1q21.1.

## Results

**Generation of Hi-C, capture-Hi-C, and haplotype phasing data.** To determine the possible effects of the 22q11.2 deletion on chromosomal interactions, we generated Hi-C contact maps for 11 human LCLs (5 patient cell lines with 22q11.2 deletion and 6 control cell lines without), with a total of 3.1 billion Hi-C contact reads of which 680 million read-pairs were of high quality and used for the downstream analyses (Supplementary Table 1). The presence and boundaries of the heterozygous 3-Mbp deletion in the patient cell lines were validated by whole-genome sequencing (Supplementary Fig. 1). To rule out an increase in overall genome instability in the patient LCLs, we carried out whole-genome sequencing at a coverage comparable to that used for the majority of samples by the 1000 Genomes Project (i.e., <10× genome-wide coverage). Almost all of our control cell lines had been part of the 1000 Genomes Project and we compared the genome-wide CNV load between our patient LCLs and LCLs used by the 1000 Genomes Project. We found no elevated genome-wide CNV burden in our patient cell lines (Supplementary Table 2).

As a means of quality control, our study included the cell line GM06990, which was the cell line used in the original Hi-C paper[20]. Inter-chromosomal contacts of GM06990 as determined by our own Hi-C data for this line (Supplementary Fig. 2a) show the same patterns of chromosomal interactions across the nucleus as in ref. [20]; i.e., small chromosomes generally have more interactions with each other than larger chromosomes with each other and many more than chromosomes in the medium size range. Specific interaction pairs, e.g., between chromosomes 17 and 19, were also replicated in our GM06990 data. Global inter-chromosomal contact maps resulting from combining all our controls and cases, respectively, again replicated these global interaction patterns (Supplementary Fig. 2b, c).

In order to be able to carry out haplotype-specific analysis of the chromatin and transcriptome data, we conducted haplotype phasing for the two related patients in our cohort, a mother and son duo with the 22q11.2 deletion, where this was made possible by combining the sequencing-based approach of statistically aided long-read haplotyping (SLRH)[21] with an analysis of Mendelian inheritance of single-nucleotide variants (SNVs). We were able to phase 1,868,316 and 1,870,948 heterozygous SNVs on the autosomes of ID00014 and ID00016, respectively (see details in Methods and Supplementary Information). Two control samples from related donors, GM12878 and GM12892, were also haplotype-phased using the same approach. For GM12892, we phased 1,929,967 autosomal SNVs and for GM12878 we phased 1,874,181 autosomal SNVs. These phased SNVs were then used to create haplotype-specific Hi-C interaction maps and to carry out allele-specific analyses of the chromatin immunoprecipitation–sequencing (ChIP-Seq) and RNA-Seq data.

Haplotype-specific Hi-C analysis requires very deep sequencing coverage of Hi-C libraries since only paired-end reads that cover at least one informative (heterozygous) SNV can be used for that purpose. To increase the Hi-C coverage for chromosome 22q to these required very deep levels, we carried out custom-designed chromosome-wide targeted capture Hi-C. We performed capture Hi-C by using a custom-designed set of 2.1 million NimbleGen oligomer capture probes representing the entire sequence of chromosome 22q and Hi-C libraries prepared using the in situ Hi-C protocol from the four cell lines for which we generated haplotype phasing data, followed by Illumina sequencing (Supplementary Table 1). With this approach, we achieved 10–16-fold levels of chromosome 22q-specific

enrichment of Hi-C paired-end reads over the standard non-capture Hi-C data (Supplementary Fig. 3).

**Normalization of Hi-C data**. Many factors, such as mappability of sequencing reads, GC content, length of the restriction enzyme fragment, etc., can lead to biases in Hi-C data. Several alternative computational approaches for Hi-C data analysis have been developed that allow for the removal of these biases from data[22–31]. When using these alternative algorithmic approaches on Hi-C data that was generated for genomes without large CNVs, such as GM12878, the resulting normalization metrics will be highly correlated[32]. For our Hi-C data, however, we needed to be certain to use a normalization method that is not thrown off at the outset by the presence of the heterozygous 3-Mbp deletion in 22q11.2.

We tested three different commonly used normalization algorithms that were developed for Hi-C data, following the rationale that the normalization methods should not change the general patterns of interaction we can see in the raw data.

We found that not all of the available normalization methods are robust for the use with Hi-C data coming from genomes with large CNVs. However, the hicpipe algorithm[22] is quite suitable for this purpose (details of the comparison between normalization methods in Supplementary Information and Supplementary Fig. 4).

**Decrease of intra-chromosomal contacts involving 22q11.2**. We observed that in 22q11DS cell lines the chromosomal contacts within the 22q11.2 deletion region are decreased significantly compared to control cell lines (Fig. 1a). Also strongly reduced are the chromosomal contacts between the 22q11.2 deletion region with the entire remainder of chromosome 22 (Fig. 1a; Supplementary Fig. 5a). This decrease is consistent with the copy number of the 22q11.2 deletion region in the patient cell lines, as all of the 22q11DS cell lines are heterozygously deleted for this region. No such decrease of chromosomal contacts that involved an extended and contiguous chromosomal region was observed that did not involve the 22q11.2 region. Likewise there was no such strong decrease of intra-chromosomal contacts over an extended and contiguous chromosomal region on any of the other autosomes (for example, chromosome 19 in Fig. 1b; Supplementary Fig. 5b).

We then investigated whether this effect of a decrease in chromosomal interactions between the 22q11.2 deletion and *cis*-contacts (i.e., within the deletion boundaries and between the deletion region and elsewhere on chromosome 22q) also holds for *trans*-contacts (i.e., for contacts between the region within the deletion boundaries and the rest of the genome). We found the *trans*-contacts involving the 22q11.2 deletion in the patient cell lines and any other chromosome also decreased compared to control cell lines (Fig. 1c; Supplementary Fig. 5c). No other regions of chromosome 22q showed such an effect and neither did any other pair of autosomes that did not include 22q (Fig. 1d; Supplementary Fig. 5d).

**Gene expression and epigenetic profiles in 22q11.2 region**. To determine the effects of the 22q11.2 deletion on gene expression patterns and chromatin marks, we performed RNA-Seq on 14 cell lines (Supplementary Table 1) and ChIP-seq for H3K27ac and H3K27me3 histone modifications in 6 cell lines, as well as ChIP-Seq for CTCF binding in 5 cell lines, respectively (Supplementary Table 3). RNA-Seq analysis showed that if a gene that is located within the boundaries of the large CNV in 22q11.2 is expressed, the level of that expression is decreased in 22q11.2DS patient lines relative to control cell lines (Fig. 1e), consistent with a previous study[8].

Differential pattern analysis of the ChIP-Seq data for H3K27ac, H3K27me3, and CTCF showed that for the majority of binding sites within the 22q11.2 deletion boundaries the binding signal decreased in 22q11.2DS patient lines compared to control cell lines (Fig. 1f–h). This phenomenon of decreased binding signals over a large chromosomal region is specific to chromosome 22q11.

**Chromosome contacts and chromatin marks in flanking regions**. The distal and proximal flanking regions of the 22q11.2 deletion are brought into close proximity to each other by the formation of the deletion breakpoint junction. We hypothesized that, since Hi-C contacts between two given regions will increase with decreasing genomic distance, the contacts between the distal and proximal deletion-flanking regions in 22q11DS patient cells would be markedly enhanced. We indeed found such stronger contacts between the deletion-flanking regions in 22q11.2DS cell lines (Fig. 1a; Supplementary Fig. 4a).

We used the haplotype phasing information and the deep capture Hi-C data for two patient and two control cell lines to generate haplotype-specific contact maps for chromosome 22q (Fig. 2a). The chromosome contacts between proximal and distal flanking regions of the 22q11.2 deletion increased on the chromosome 22q with the deletion when compared with the intact chromosome 22q within the same patient cell line (Fig. 2). The chromosome contacts between proximal and distal deletion-flanking regions in 22q11.2 on the intact chromosome in the patient cell lines were not affected compared to the chromosomes in the controls (Fig. 2).

Following this observation, we wanted to examine whether there is an effect on the chromatin marks concurrent to these increased chromosomal contact patterns. To do so, we performed enrichment analysis in our ChIP-Seq data to detect genomic regions that were enriched with significantly differential signals of H3K27ac, H3K27me3, and CTCF between patients and controls in 500-kbp bins. We found that the deletion-flanking regions were significantly enriched for differences in H3K27ac and H3K27me3 signal (Fig. 3a, b). For CTCF, only the distal deletion-flanking region was significantly enriched (Fisher's exact test $p$ value = 2.53e−06) (Fig. 3c).

We then determined which individual binding sites within the differential 500-kbp bins contributed to the differential signal. Within the proximal deletion-flanking region, from 18 to 18.5 Mbp, we found that 5 out of the 24 sites with ChIP-Seq signal for H3K27ac and 6 out of the 15 sites with ChIP-Seq signal for H3K27me3 showed significantly differential binding (Fisher's exact test $p$ value = 0.0075 for H3K27ac and $p$ value = 5.67e−05 for H3K27me3, respectively). Interestingly, for H3K27ac all of the five sites with significant differential binding were bound less strongly while for the same region for H3K27me3 all of the six sites with significant differential binding were bound more strongly in the 22q11DS patient lines. For the distal region, from 23 to 23.5 Mbp, 5 out of the 13 sites with H3K27ac binding and 11 out of the 30 sites with H3K27me3 binding showed significantly differential binding (Fisher's exact test $p$ value = 0.0004 for H3K27ac and $p$ value = 1.26e−07 for H3K27me3, respectively). Again we observed the reciprocity between significantly differential changes for the two different histone marks: 4 out of the 5 sites for H3K27ac gave a less strong signal while all of the 11 of such sites for H3K27me3 showed a stronger signal in the 22q11.2DS patient cell lines.

Such reciprocity in signal strengths between these two histone marks is a general feature of their principle of action and indicates that the observed changes in chromatin marks are of physiological relevance.

**Haplotype-specific A/B compartments and topological domains**. Previous studies of cell lines without large CNVs have

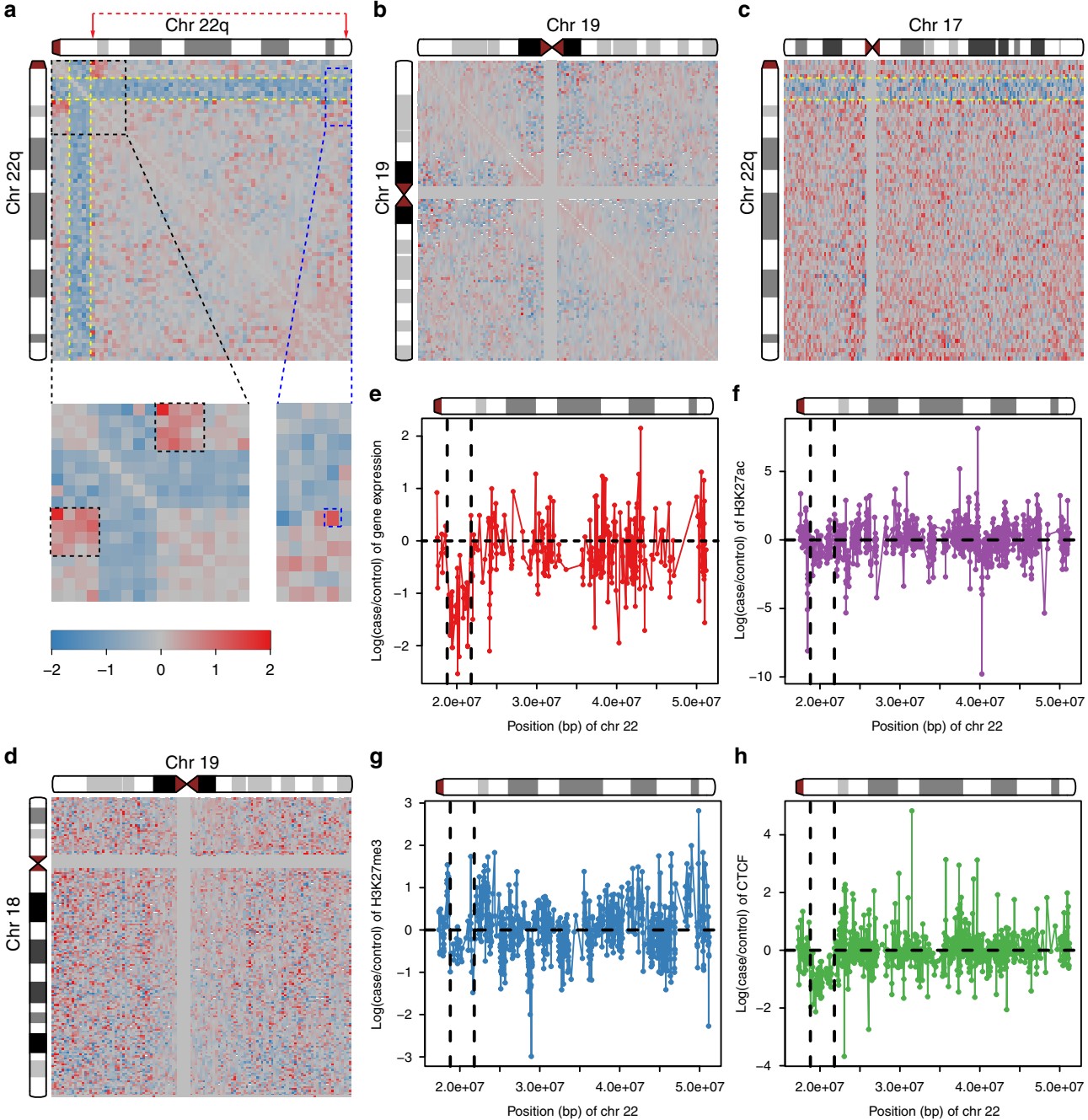

**Fig. 1** Effects of the 22q11.2 deletion on chromosome interactions, gene expression and chromatin marks. **a–d** Each pixel in the heatmaps represents the intra- or inter-chromosomal contact frequency in Hi-C data from 22q11.2del cell lines ($n = 5$) versus control cell lines ($n = 6$) for a 500-kbp region. Yellow dashed lines indicate the 3-Mbp deletion on chromosome 22q. The color scale goes from −2 (blue) to 0 (gray) to 2 (red). **a** Fold change of cis-contacts along chromosome 22 in 22q11.2del versus control cell lines. Black boxes indicate increased contacts between the deletion-flanking regions in 22q11.2del cell lines. Blue box: the signal for increased contacts between the centromere–distal deletion-flanking region and the telomeric end of chromosome 22q (red arrows and dashed red line indicate the corresponding chromosome folding event). **b** Lack of intra-chromosomal fold change of contacts for chromosome 19. **c** Fold change of inter-chromosomal contacts between chromosome 22 and chromosome 17. **d** Lack of inter-chromosomal fold change of contacts between chromosome 18 and chromosome 19. **e** Log2-transformed fold change of gene expression for genes on chromosome 22q in RNA-Seq data from 22q11.2del ($n = 5$) versus control ($n = 9$) cell lines. Each point represents a gene. **f–h** Log2-transformed fold change in ChIP-Seq signals in 22q11.2del versus control cell lines. **f** H3K27ac histone modifications ($n = 5$ for 22a11.2del and $n = 4$ for control cell lines). **g** H3K27me3 histone modifications ($n = 5$ for 22a11.2del and $n = 4$ for control cell lines). **h** CTCF-binding sites ($n = 4$ for 22a11.2del and $n = 3$ for control cell lines). Black vertical dashed lines indicate the 3-Mbp deletion on chromosome 22q in **e–h**

revealed that there is a level of organization in the cell's nucleus where the chromatin is partitioned into two states, termed A and B compartments, which broadly correspond to large regions that are overall active and overall inactive, respectively[20]. These A/B compartments can be derived from Hi-C data. We sought to determine whether the 22q11.2 deletion might lead to changes in the A/B compartments. We computed the haplotype-specific A/B compartments for the four cell lines for which both deep capture

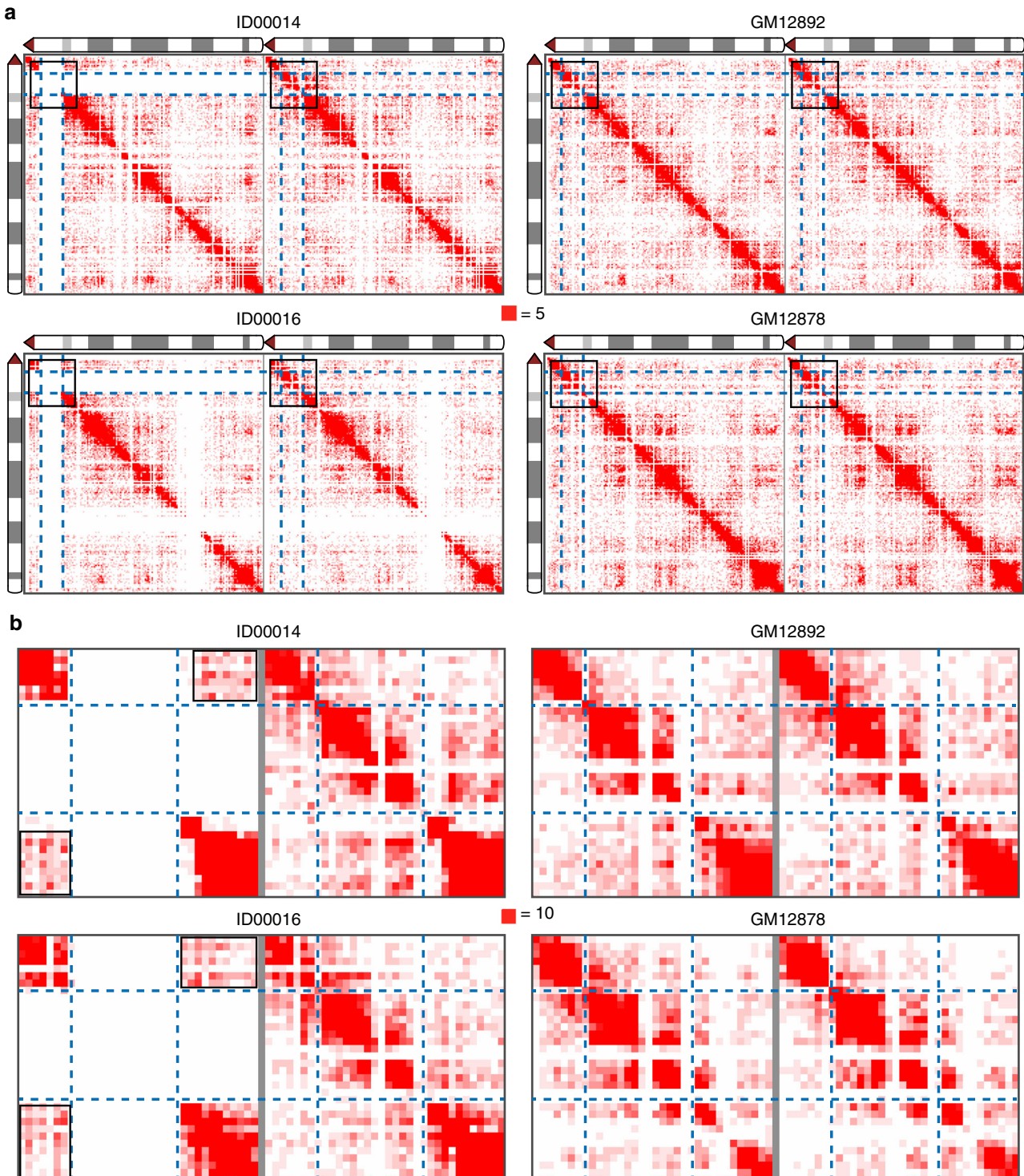

**Fig. 2** Haplotype-specific intra-chromosomal contacts of chromosome 22q. **a** Contacts within each homologous chromosome 22q for patients (ID00014 and ID00016) and controls (GM12878 and GM12892) at 200 kbp resolution. The chromosome 22q with 22q11.2 deletion is shown on the left for the two patient cell lines. Intensity of contacts is represented by a scale from 0 (white) to 5 (deep red). The gap in signal for the region from 39 to 42 Mbp in ID00016 is caused by a very low density of heterozygous SNVs. **b** Zoomed-in view of the regions in black boxes in **a**. The color scale goes from 0 (white) to 10 (red)

Hi-C and genome haplotype phasing data are available (Fig. 3d). We did not observe differences in the A/B compartments between cell lines from patients and from controls and neither between the two homologous chromosomes 22q in any individual cell line, patient, or control. Our results indicated that A/B compartments of the homologous chromosome with the 22q11.2 deletion were not affected by the deletion.

Topological domains (also known as topologically associating domains (TADs)) are a megabase-sized structural feature of the genome organization that is constituted of highly self-interacting

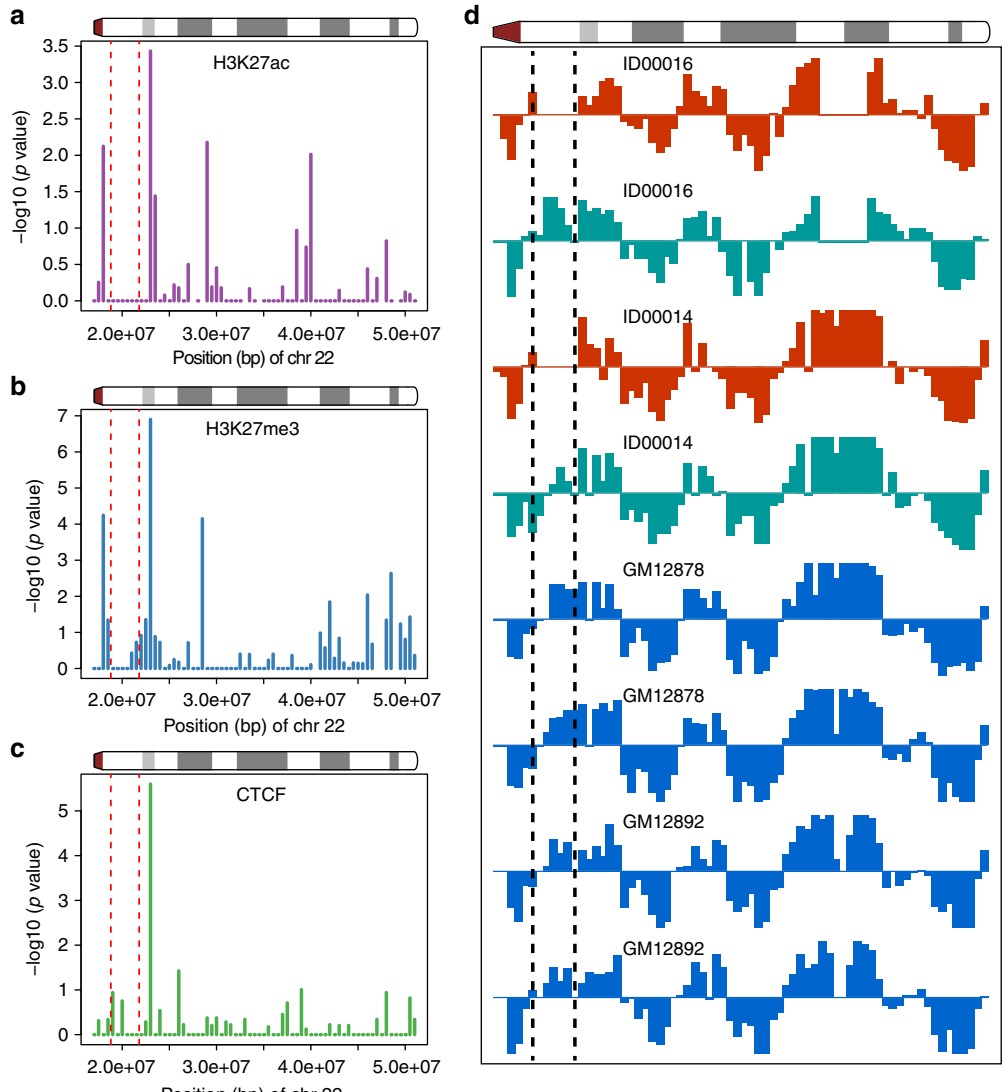

**Fig. 3** Distribution of significantly enriched differential ChIP-Seq signals and haplotype-specific A/B compartments. **a** H3K27ac histone marks ($n = 5$ for 22a11.2del and $n = 4$ for control cell lines). **b** H3K27me3 histone marks ($n = 5$ for 22a11.2del and $n = 4$ for control cell lines). **c** CTCF-binding sites ($n = 4$ for 22q11.2del and $n = 3$ for control cell lines). **d** A/B compartments for each homologous chromosome 22q. Each bar represents a 500-kbp bin. Red and black dashed lines mark the boundaries of the 3-Mbp deletion in 22q11.2. Shown in **d** is the first eigenvector for the principal component analysis of the chromosomal contact matrix. *x* Axis: position on chromosome 22q. *y* Axis: value of the first eigenvector. The signal from the homologous chromosome 22q with 22q11.2 deletion and the intact chromosome 22q are shown in red and turquoise, respectively, for patients (ID00016 and ID00014). Signals from the homologous chromosomes 22q of the controls are in blue

chromosome regions[33]. We calculated the haplotype-specific topological domains on the four cell lines with both capture Hi-C and haplotype phasing data (Fig. 4, Supplementary Figs. 6-7). Although there were variations in the calling of topological domains across individuals and between the homologous chromosomes, the direction indices were highly consistent between the two homologous chromosomes in the controls (Supplementary Fig. 6c-d, Supplementary Fig. 7) as well as in the patients (Fig. 4, Supplementary Fig. 6a-b), suggesting no changes of topological domains on the homologous chromosome with the 22q11.2 deletion. In spite of the increased contacts between the flanking regions of the deletion, they were not strong enough to incur fusion of the topological domains proximal and distal of the 22q11.2 deletion. However, the increased contacts between the flanking regions of the deletion extended to multiple deletion-distal topological domains.

**Change in *cis*-contacts of distal deletion-flanking region.** Based on the above findings that the genomic region distal of the 22q11.2 deletion was affected on different levels of molecular regulation in 22q11.2DS patient cell lines, we next sought to investigate whether the presence of the 22q11.2 deletion also affected the chromosomal *cis*-contacts of the distal deletion-flanking region with any other region on chromosome 22q. To do so, we analyzed our Hi-C data by calculating the fold change of contacts between the 21.5 and 22-Mbp window, which is situated right distal of the 22q11.2 deletion, and all the other 1-Mbp-sized regions on chromosome 22q. We observed that region 17–18 Mbp, i.e., the breakpoint-proximal region right proximal of the deletion, showed the largest fold change (2.04) of contact with region 21.5–22 Mbp between 22q11DS and control cell lines. This was as expected given that these chromosomal regions were brought into close proximity to each other by the 22q11.2 deletion (Fig. 1a).

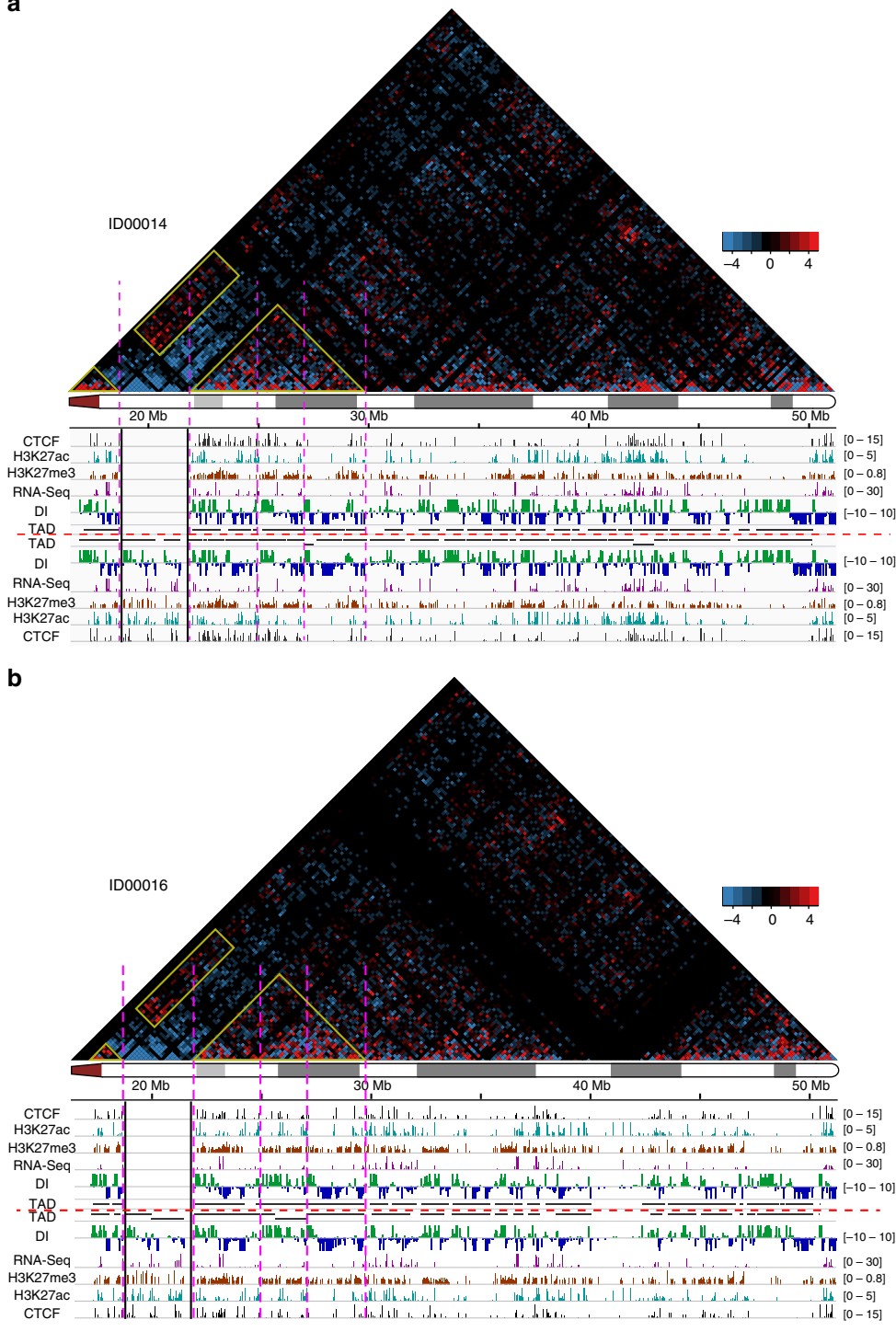

**Fig. 4** Differential chromosome contacts for the two chromosomes 22q and haplotype-specific topological domains for two patient cell lines. **a** Patient ID00014, **b** Patient ID00016. Heatmaps with bin size 200 kbp are showing the intra-chromosomal contacts on the homologous chromosome 22q with the 22q11.2 deletion minus the intra-chromosomal contacts on the intact chromosome 22q. The color scale of the heatmap goes from −5 (blue) to 5 (red) with 0 represented by black. Yellow rectangles mark the increased contacts between the topological domains proximal and distal of the 22q11.2 region on the homologous chromosome 22q with the deletion (the topological domain on the proximal side of the deletion increases its contact not just with the topological domain immediately distal of the deletion, as predicted, but also with the two following topological domains). Yellow triangles and pink dashed lines mark the topological domains involved in the increased contacts. Black lines mark the boundaries and extent of the 22q11.2 deletion. Signal tracks for CTCF binding, histone marks, RNA-Seq, direction indices (DIs), and topological domains are shown twice, for the chromosome 22q with deletion (above dashed red line) and for the intact chromosome 22q (below dashed red line). Direction indices (DIs) are calculated at 40-kbp resolution for each homologous chromosome 22q. Haplotype-specific gene expression, histone modifications, and CTCF binding, respectively, are shown after normalizing the reads to 10 million read pairs. RNA-Seq and ChIP-Seq read-count bars within the boundaries of the deletion but from the intact chromosome 22q are shown for loci where there are phased SNVs in GM12878

The second largest fold change (1.96) of *cis*-contacts involving region 21.5–22 Mbp was for contacts with region 50–51 Mbp, i.e., toward the very telomeric end of chromosome 22q (Fig. 1a). Intriguingly, we also observed strong positive correlation between region 22–22.5 Mbp and region 50–50.5 Mbp for CTCF binding (Pearson's $r = 0.933$, $p$ value = 0.02) (Supplementary Fig. 8a) and between region 22–22.5 Mbp and region 50.5–51 Mbp for H3K27ac enrichment (Pearson's $r = 0.811$, $p$ value < 0.05) (Supplementary Fig. 8b). Furthermore, there was weak correlation between region 22–22.5 Mbp and 50.5–51 Mbp for H3K27me3 enrichment (Pearson's $r = 0.74$, $p$ value = 0.090) (Supplementary Fig. 8c). Given that both regions 21.5–22 and 22–22.5 Mbp are in the distal flanking region of the 22q11.2 deletion, our results indicate that increased chromosomal contacts between the distal deletion-flanking region and the telomeric region 50–51 Mbp may be associated with the differential changes of histone modifications and CTCF binding that we found to be in correlation between these two regions.

To validate the increased chromosomal contact between the deletion-distal region and the telomeric end of 22q, we performed three-dimensional fluorescence in situ hybridization (3D FISH) on 5 cell lines with the 22q11.2 deletion and on 6 control cell lines, using two FISH probes specific to loci in region 21.8–22.5 Mbp (RP11–47L18: chr22:21,931,796–22,118,344) and in region 50–51 Mbp (RP11–125K3: chr22:50,149,996–50,287,311), respectively. FISH showed that the distal flanking region of the 22q11.2 deletion is significantly closer to region 50–51 Mbp in nuclear space in 22q11DS cell lines than in control cell lines (*t* test *p* value 0.008) (Fig. 5; Supplementary Fig. 9). Taken together, our results strongly indicate that the 22q11.2 deletion causes conformational changes on several levels on chromosome 22q.

**Global changes of inter-chromosomal contact patterns**. To explore the effect of the 22q11.2 deletion on *trans*-contacts, i.e., between any non-homologous autosomes in the nucleus of 22q11.2DS patient lines, we analyzed our Hi-C data for significantly different *trans*-contacts between 22q11DS and control cell lines. On the genome-wide level, we found 272 *trans*-contacts with a Fisher's exact test *p* value of <0.0001 (Fig. 6a). Interestingly the majority of these chromosomal *trans*-contacts did not involve chromosome 22q as one of the interacting partners. Notably, 56 of these inter-chromosomal contact signals are among the top 5% of the strongest *trans*-contacts (Fig. 6b). This enrichment of significantly different genome-wide chromosomal *trans*-contacts within the strongest *trans*-contacts is again highly statistically significant (Fisher's exact test *p* value < 2.2e−16). We found far fewer significantly different genome-wide chromosomal *trans*-contacts by randomly swapping the assignment of 22q11.2 deletion and control status across our Hi-C data sets, by comparing within control cell lines and by comparing within 22q11.2del cell lines (Supplementary Table 4). None of these swapping analyses achieved the same enrichment. This indicates that a relevant amount of the significantly different genome-wide chromosomal *trans*-contacts is not due to random chromosomal motion or to as-of-yet unknown factors such as cell culture variations across the LCLs. Rather, our analysis points to a genome-wide disturbance of the network of chromosomal *trans*-contacts that is at least in part attributable to the presence of the 22q11.2 deletion on chromosome 22q.

**Global changes of gene expression patterns**. To investigate the global effect of the 22q11.2 deletion on gene expression, we performed differential expression analysis between the 22q11.2DS and control cell lines. Of the 11,374 genes with detectable levels of expression (fragments per kilobase of transcript per million mapped reads (FPKM) > 0.5), 1610 genes showed significantly

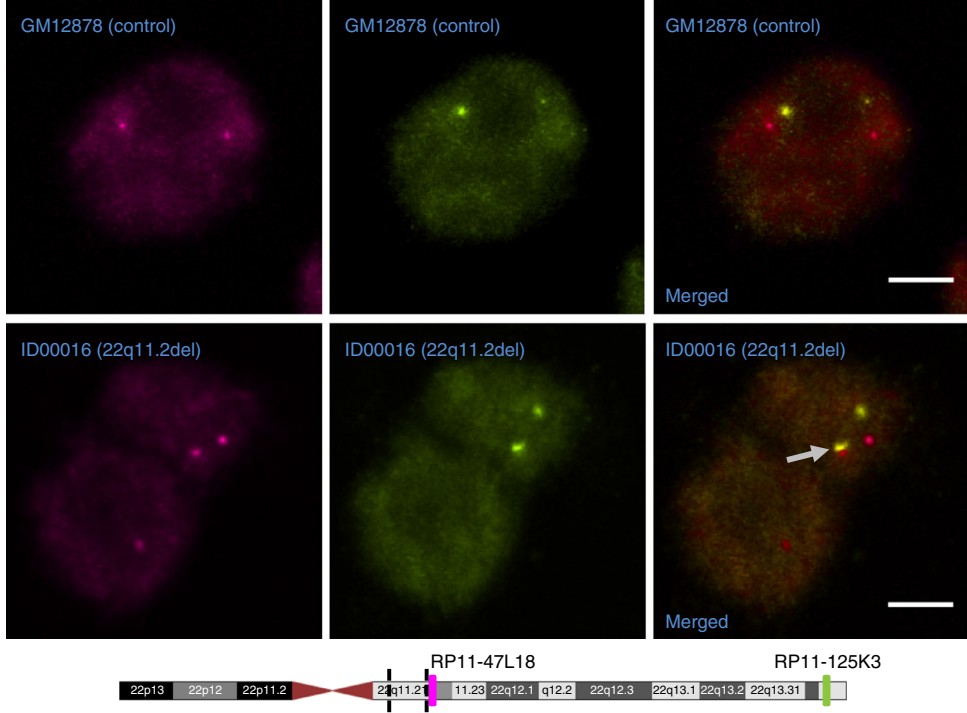

**Fig. 5** Examples of 3D FISH visualization of intra-chromosomal interaction changes. The regions for which Hi-C predicted changes in interaction, regions 21.8–22.5 and 50–51 Mbp on chromosome 22q, were visualized by 3D DNA FISH using BAC probes RP11-47L18 and RP11-125K3, labeled with biotin (magenta) or digoxigenin (green). Arrow: an example for the magenta and green FISH probes in close proximity in a cell carrying the 22q11.2 deletion. Scale bars are 5 µm. Magenta and green bars on chromosome 22q indicate the locations of the biotin and digoxigenin FISH probes, respectively. Black dashed lines indicate the position of the 3-Mbp deletion

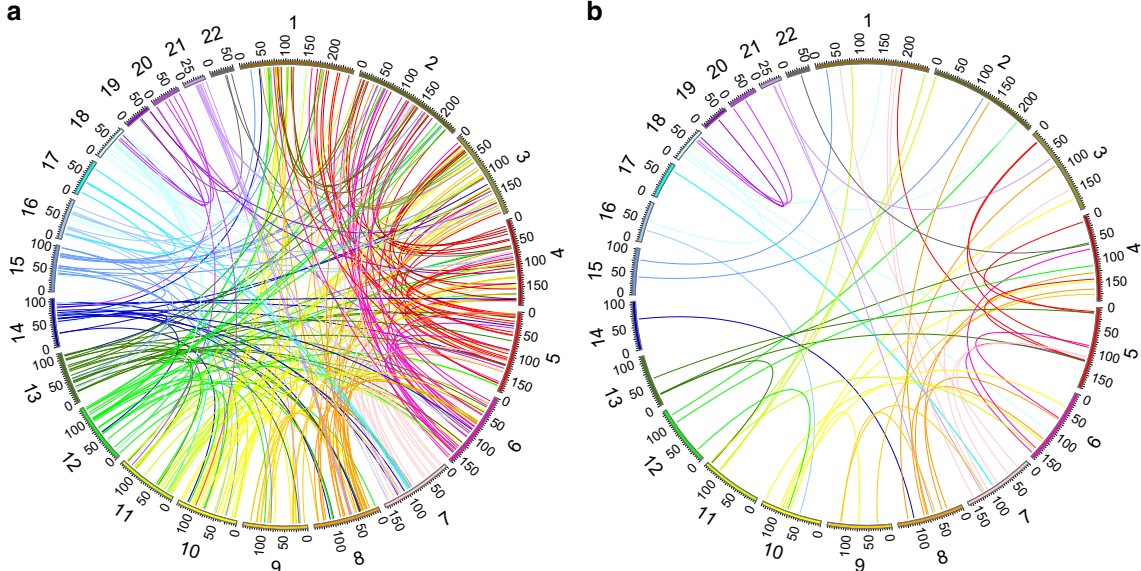

**Fig. 6** Genome-wide inter-chromosomal contact changes in 22q11.2del versus control cell lines. **a** Circos plot of the inter-chromosomal contacts exhibiting differential interaction in 22q11.2del ($n = 5$) versus control ($n = 6$) cell lines at significance level of Student's $t$ test $p$ value $= 0.0001$. **b** Circos plot of the inter-chromosomal contacts exhibiting differential interaction in 22q11.2del ($n = 5$) versus control ($n = 6$) cell lines at significance level of Student's $t$ test $p$ value $= 0.0001$ and showing only the top 5% strongest inter-chromosomal contacts. The circle displays all autosome-sized scaffolds and each line represents an inter-chromosomal contact change in **a**, **b**

differential expression (false discovery rate (FDR) < 0.05). Gene ontology analysis indicated that these differentially expressed genes are enriched for genes involved in mitochondrial pathways such as the respiratory chain ($n = 32$, modified Fisher's exact test $p$ value $= 2.96e-11$). KEGG pathway analysis demonstrated enrichment of genes involved in oxidative phosphorylation and neurodegenerative diseases, such as Parkinson's disease (Fig. 7a). Earlier studies have noted that there are several genes related to mitochondrial function that are located within the 22q11 deletion boundaries[34], therefore the pathways associated with this cellular function would have a high likelihood to be affected by the change in copy number. The enrichment for pathways related to neurodegenerative disorder is unexpected in LCLs but notable, as an association between 22q11DS and Parkinson's Disorder has been reported previously[35–37]. This may indicate that the LCL cell culture may in certain instances be of relevance for the molecular study of 22q11DS even beyond the fundamental and general levels.

We also carried out genome-wide analysis to determine whether there are entire genomic regions that are enriched for differentially expressed genes. As expected, the most significant signals for this analysis were located in the 22q11.2 region (Fig. 7b; Supplementary Fig. 10). No other regions genome wide achieved FDR-corrected significance (Supplementary Fig. 10).

**Correlation between histone modification and gene expression.** To examine whether the gene expression changes are consistent with the histone modification changes, we assigned the H3K27ac and H3K27me3 peaks to their nearest genes based on the distance to their transcription start sites (TSSs), and for each gene, we only retained the closest peak for both histone marks. We observed that significantly upregulated genes in 22q11.2DS cell lines exhibited significantly higher fold change of H3K27ac enrichment (permutation test $p$ value $= 0.0140$) and significantly lower fold change of H3K27me3 enrichment (permutation test $p$ value $= 0.0018$) than genes non-significantly upregulated in 22q11.2DS cell lines (Fig. 7c). Consistently, significantly downregulated genes in 22q11.2DS cell lines exhibited significantly lower fold change

of H3K27ac enrichment (permutation test $p$ value $= 0.0002$) but significantly higher fold change of H3K27me3 enrichment (permutation test $p$ value $= 0.0227$) than genes non-significantly downregulated in 22q11.2DS cell lines (Fig. 7c).

Moreover, genes whose TSSs showed significantly upregulated binding by H3K27ac in 22q11DS cell lines exhibited significantly higher fold change of expression (permutation test $p$ value $< 2.2e-16$) than those with non-significantly upregulated binding, while those TSSs showing significantly downregulated binding exhibited a significantly lower fold change of expression (permutation test $p$ value $= 0.0202$) than those with non-significantly downregulated binding (Fig. 7d). Consistently, genes whose TSSs showed significantly upregulated binding by H3K27me3 in 22q11DS showed significantly lower fold change of expression (permutation test $p$ value $= 0.0037$), whereas those with significantly downregulated binding exhibited significantly higher fold change of expression (permutation test $p$ value $= 0.0027$) between 22q11DS and control cell lines (Fig. 7d).

We also calculated the correlation between gene expression and H3K27ac-binding affinity across the individuals for all the genes with TSSs bound by H3K27ac. In line with the above results, we found significantly higher Pearson's correlation coefficients than the coefficients obtained from permutations (Wilcoxon rank sum test $p$ value $< 2.2e-16$) (Supplementary Fig. 11a). This difference was even more significant when we included the differential expressed genes (absolute fold change >2) only (Wilcoxon rank sum test $p$ value $< 2.2e-16$) or the differential H3K27ac-bound genes only into our analysis (Wilcoxon rank sum test $p$ value $< 2.2e-16$) (Supplementary Fig. 11a). We also observed significantly higher Pearson's correlation coefficients between gene expression and H3K27me3 binding than the coefficients obtained from permutations for the differential expressed genes (absolute fold change >2) only (Wilcoxon rank sum test $p$ value $= 9.72e-04$) and the differential H3K27me3-bound genes only (Wilcoxon rank sum test $p$ value $= 3.71e-06$) (Supplementary Fig. 11b). Together, our results demonstrate that gene expression changes are associated with histone modification changes in

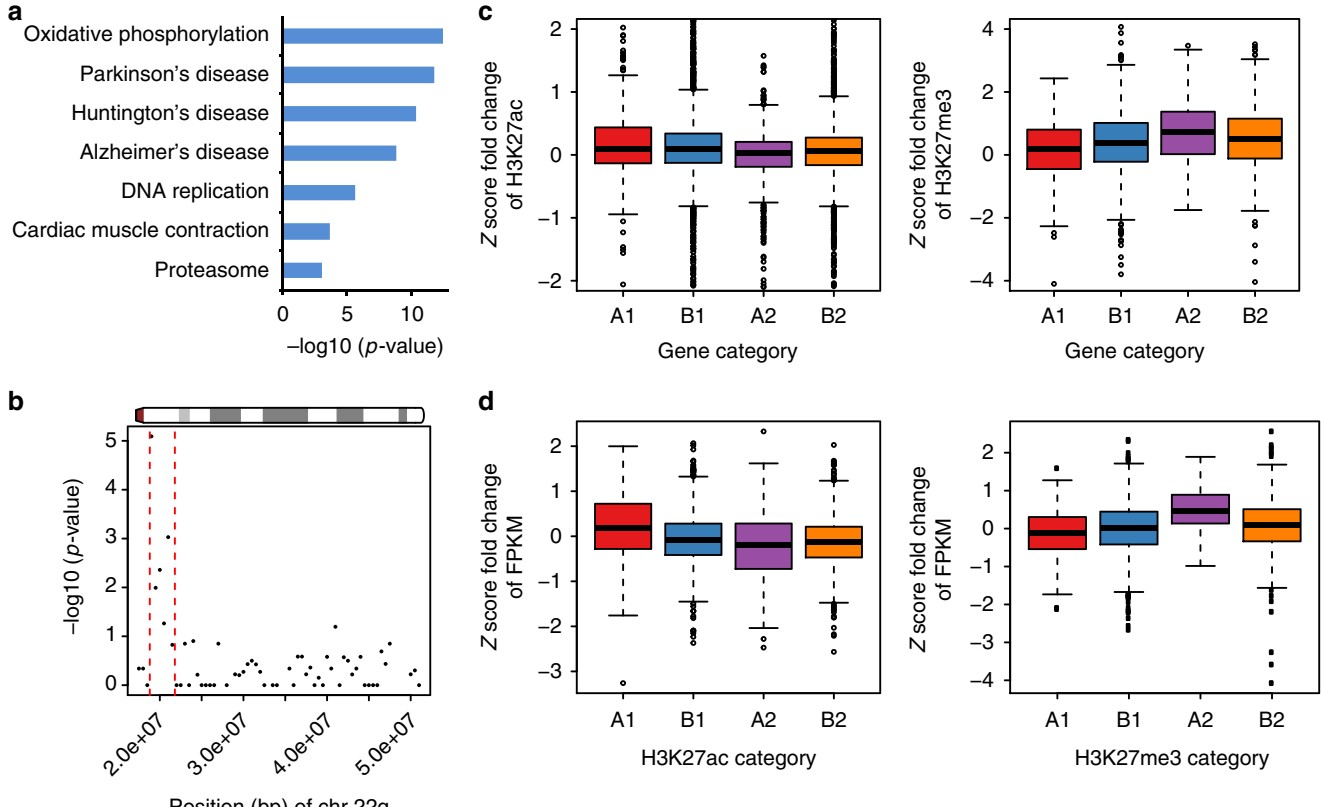

**Fig. 7** Differential expression analysis and correlation between gene expression and epigenetic profiles. **a** KEGG pathway analysis for genes differentially expressed between 22q11.2del ($n = 5$) and control ($n = 9$) cell lines. Shown are pathways with modified Fisher's exact test $p$ value $< 9.05e{-}04$. **b** Enrichment analysis for chromosome 22q for differentially expressed genes. Red dashed lines mark the boundaries of 22q11.2del region. **c** Differentially expressed genes correlated with different H3K27ac and H3K27me3 signal strengths in 22q11.2del versus control cell lines. A1 represents genes significantly upregulated in 22q11.2del cell lines while B1 represents non-significantly upregulated genes; A2 represents genes significantly downregulated in 22q11.2del cell lines while B2 represents non-significantly downregulated genes. The $y$ axis shows the $Z$-score transformed fold change of H3K27ac signals (left) or H3K27me3 signals (right) between 22q11.2del ($n = 5$) and control ($n = 4$) cell lines. **d** Differential H3K27ac and H3K27me3 signal strengths exhibited correlation with differential gene expression between 22q11.2del and control cell lines. A1 represents genes whose TSSs were significantly more marked by H3K27ac (left) or by H3K27me3 (right) in 22q11.2del cell lines. B1 represents genes with TSSs non-significantly more marked by H3K27ac (left) or by H3K27me3 (right). A2 represents genes whose TSSs were significantly less marked by H3K27ac (left) or by H3K27me3 (right) in 22q11.2del cell lines. B2 represents genes with TSSs non-significantly less marked by H3K27ac (left) or by H3K27me3 (right). The $y$ axis shows the $Z$-score transformed fold change of genes' FPKM in 22q11.2del ($n = 5$) versus control ($n = 9$) cell lines. In **c**, **d**, box represents quartiles, centre line denotes 50th percentile, and whiskers extend to the most extreme data points within 1.5 times of the interquartile range

22q11DS cell lines compared with control cell lines and that this phenomenon occurs across the entire genome in cells with the deletion in 22q11.2.

**Effects of 1q21.1 deletion on chromosome folding**. To explore whether large CNVs other than the one on chromosome 22q11.2 can lead to changes in the patterns of chromosome folding, we performed Hi-C on two LCLs with a heterozygous deletion of approximately 1.35 Mbp in size on chromosome 1q21.1 (1q21.1del). This deletion is strongly associated with the development of schizophrenia[38–40]. Similar to what we had found for the 22q11.2 deletion, we observed that both *cis*- and *trans*-contacts between the 1q21.1 deletion regions and other regions were decreased in 1q21.1del cell lines relative to control cell lines (Fig. 8a, b). As observed in the 22q11DS cell lines, in the 1q21.2del cell lines there was an increase of intra-chromosomal contacts between the regions directly flanking the main CNV of 1q21.1 (Fig. 8a). Global changes of inter-chromosomal contacts were also observed (Supplementary Fig. 12), again consistent with the inter-chromosomal contact changes in 22q11.2 LCLs. Taken together, cells with the large deletion CNV on chromosome

1q21.1 exhibited similar effects on chromosomal *cis*-contacts and *trans*-contacts as found in cells with the large deletion CNV on chromosome 22q11.2, which points toward our findings in 22q11DS being generalizable across large neuropsychiatric CNVs.

**Discussion**
Large CNVs are an important feature of the genetic architecture of several major neurodevelopmental psychiatric disorders as well as of conditions involving aberrant morphology of many organ systems. Their effects on the level of phenotype are complex and the molecular mechanisms mediating these effects are very incompletely understood. While it is still a good assumption that a considerable portion of these mechanisms are a direct consequence of the copy number change of the genes within a given CNV's boundaries and the resulting changes in expression levels for these genes, it also seems plausible to investigate whether additional levels of complexity exist regarding the effects of a large CNV across multiple layers of the control of gene activity. This plausibility stems from several observations and lines of reasoning. There are large numbers of genes affected genome wide and far distal from the main CNV. This leads us to consider

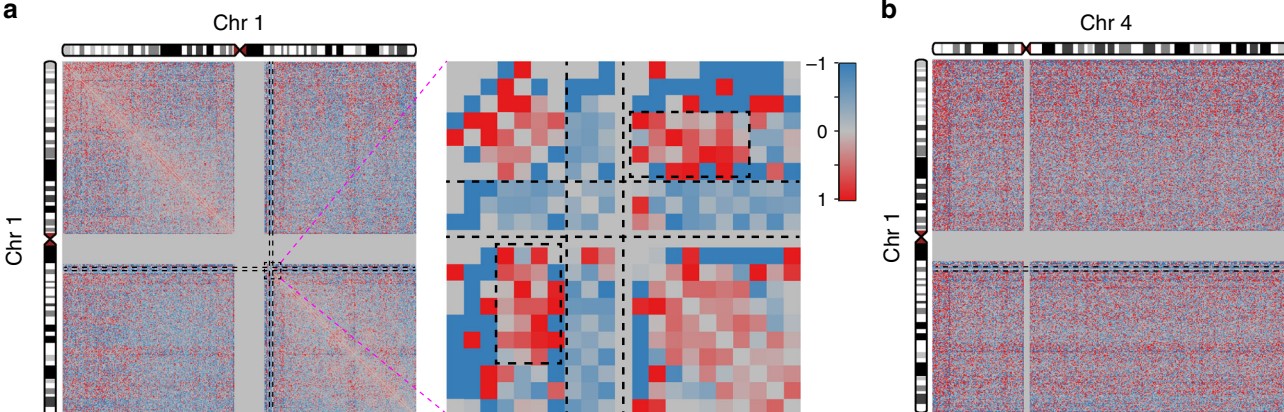

**Fig. 8** Effect of the large deletion CNV on 1q21.1 on chromosome conformation. **a** Fold change of *cis*-contacts of chromosome 1 in 1q21.1del (*n* = 2) versus control (*n* = 6) cell lines. Black boxes mark the regions of increased contacts between the proximal and distal flanking regions of the 1q21.1 deletion. **b** Fold change of *trans*-contacts between chromosome 1 and 4 in 1q21.1del (*n* = 2) versus control (*n* = 6) cell lines. In **a**, **b**, black dashed lines mark the boundaries of the 1q21.1 deletion. Each cell in the heatmap represents the chromosomal contact level between two 500 kbp regions. The color scale goes from −1 (blue) to 0 (gray) to 1 (red)

the basic principles that govern organizational features of the nucleus and the chromatin such as regulatory domains being bounded by protein factors that recognize binding sites in the DNA sequence, which in turn could be affected (i.e., deleted or duplicated) by a CNV, or long-range chromosome contacts being also influenced by the length of the involved chromosomes (and that length being changed by large CNVs). Finally, there have been several recent reports about mutations in chromatin remodeling genes in the context of neuropsychiatric disorders pointing to the importance of proper molecular management on the epigenomic level in these conditions.

Here we studied the effects of the important large CNV on chromosome 22q11.2, and in a more limited fashion the effects of the large CNV on chromosome 1q21.2, on chromatin conformation including long-range chromosome contacts and domain formation, epigenetic profiles, and gene expression. We developed an approach that combined genome phasing and targeted fragment enrichment for very deep chromosome-wide Hi-C, which makes it possible to investigate the effects of the 22q11.2 deletion in a haplotype-specific fashion. We observed multiple effects of the large CNVs on long-range chromosome contacts, chromatin domains, and epigenetic profiles as well as on gene expression. More specifically, we found increased contacts between distal and proximal flanking regions of the 22q11.2 deletion in 22q11DS cell lines in contrast with control cell lines. Our haplotype-specific analyses of chromosomal interactions demonstrated that these increased contacts only occurred on the chromosome 22q carrying the deletion. Interestingly, both distal and proximal flanking regions of the 22q11.2 deletion were enriched with differential signals for the histone marks H3K27ac and H3K27me3 while only the distal flanking regions were enriched with differentially binding sites of CTCF. At present, the reason for this discrepancy is not known. We note that only the deletion-distal flanking region is engaging in increased intra-chromosomal interactions with the telomeric end of chromosome 22q but the available data are not sufficient to conclude that differential binding of CTCF is causally involved in this phenomenon.

We did not observe the end of chromosome 22 (50–51 Mbp) to be enriched with differentially expressed genes (2 out of the 24 genes) despite the Hi-C finding, validated by FISH, of significantly increased contacts with the distal flanking region of the 22q11.2 deletion in 22q11.2DS cell lines. Allele-specific expression analysis did not provide evidence for the existence of differential expression between the intact chromosome 22q and the chromosome with the 22q11.2 deletion (Supplementary Fig. 13),

outside the CNV boundaries. Taken together, our results indicate that chromosome conformation changes caused by the 22q11.2 deletion contribute to gene expression changes and epigenetic profile changes between 22q11.2DS and control cell lines, but not always in a linear and clearly deterministic fashion.

However, we did observe significant positive correlation between H3K27ac changes and gene expression changes but negative correlation between H3K27me3 and gene expression changes on the genome-wide level. These findings demonstrated that the epigenetic profiles were reshaped genome wide in correlation with extensive gene expression changes in 22q11.2DS cell lines.

Our pathway enrichment analysis of differentially expressed genes revealed candidate pathways that might be associated with various clinical phenotypes of 22q11DS, such as cardiac symptoms and neurodegenerative disorders (Fig. 7a), in addition to general molecular pathways, some with obvious potential association with 22q11DS such as mitochondrial energy metabolism. It is important to remain cautious when interpreting these disease-specific results as they are based on data from a cell type that is not directly relevant to the organ systems in question. At the same time, many genes will of course be relevant in more than one cell type and organ, and their expression levels may be regulated by similar molecular mechanisms across cell types. With the proper caution in place, the findings about potential relevant pathways may be considered as an unexpected side product of this study that may point toward additional routes for further analysis. In any case, the pathway analyses do not affect the findings about effects on chromatin states and chromosome folding in 22q11DS.

Our study does not address the question of what the molecular causes for phenotypic variance between individual carriers of the 22q11.2 deletion might be. This is a highly important question but one that, given the vast number in each individual genome of potential genetic modifiers, that together with environmental factors are the prime suspects for such variance, will need far larger cohort sizes of cell lines modeling the tissues where such variance is observed. Such cohorts are currently being assembled and will be the basis of highly interesting work in the coming years[41].

The effects of decreasing *cis*- and *trans*-contacts and the increased chromosomal contacts between distal and proximal deletion-flanking regions in 22q11Ds cell lines were also observed in cell lines with the 1q21.1 deletion. That such similar results were obtained on this level of observation for both the 22q11.2 deletion and the 1q21.1 deletion could suggest that at least some

of the changes caused by a large deletion CNV are generalizable for such chromosomal aberrations instead of specific to the 22q11.2 deletion.

As a cautionary note on the technical level, we demonstrated here that for genomes with a large deletion CNV the appropriate normalization methods for Hi-C data have to be chosen with great care to avoid false findings. For instance, we would have reached the conclusion that the chromosomal contacts within the deletion regions are not decreased in cell lines with deletion compared with control cell lines if the hiclib software package[23] had been used for normalization of Hi-C data, instead of the hicpipe package[22].

Lastly, we were able to generate haplotype-specific maps of chromosomal contacts by developing an approach that combines genome-sequencing-based phasing information and chromosome-wide capture of Hi-C fragments before sequencing. These haplotype-specific contact analyses enabled us to distinguish between the intact chromosome 22q and the chromosome 22q with deletion. Based on this, we derived chromosomal A/B compartments and topological domains for both homologous chromosomes 22q separately. There was no switch of A/B compartments on the chromosome 22q with the 22q11.2 deletion compared to the intact chromosome 22q in the patients. There also was no evidence for newly fused topological domains or changes of topological domain boundaries on the chromosome 22q with the 22q11.2 deletion. Earlier, Lupiáñez et al.[42], using the 4C assay, demonstrated in a mouse model as well as in human lines from patients with rare malformations of the limbs, that large structural changes in the genomic sequence on mouse chromosome 1 and human chromosome 2 can affect the topological domain architecture that is situated directly on top of the sequence change. However, our results did not show comparable effects of the 22q11.2 deletion. Further studies will determine which factors, such as, for example, the size of a given CNV or its position relative to topological domain-defining sequence elements, might determine whether topological domains are altered or not. Interestingly, although the topological domains on or nearby the 22q11.2 deletion themselves were not altered (rather, they were either deleted entirely or left unchanged), there was an unexpected pattern of increase in contact between the topological domains flanking the deletion, where the proximal topological domain not only increased its frequency of interaction with the nearest distal topological domain but also with the two topological domains following that.

In summary, we found multiple effects of the large deletion CNV on chromosome 22q11.2 on long-range chromosome contacts, chromatin organization, epigenetic profiles, and gene expression. Extensive changes on these levels caused by the 22q11.2 deletion are global and seem to be rippling along the entire chromosome carrying the deletion as well as across the entire nucleus rather than being confined to the deletion region only. Such effects had never been shown before outside of cancer cell lines[43]. Furthermore, in contrast to the findings in cancer cells we used a larger cohort of individual patient cell lines for most of our analyses, with all lines carrying only one main large CNV that is clearly and strongly associated in a causative manner with a neurodevelopmental phenotype. The earlier study in cancer used only two cancer cell lines that each carried multiple large CNVs which could also have been a consequence rather than a cause of the disease phenotype. This makes it much more likely that the higher-order effects of the large CNVs that we observed may be contributing to the molecular etiology of the developmental disorders in question, a point which is further strengthened by another recent paper[9], where the authors described studying the effects of large CNVs on chromosome 16p11 on chromosome interactions. The large CNVs on 16p11 are almost as strongly associated with neurodevelopmental disorders as the large CNV on 22q11.2. While the study on the 16p11 CNVs used a somewhat smaller number of cell lines than our study and also used 4C

as a method of discovery, which is, unlike the Hi-C method used by us, not able to detect changes in a global and unbiased fashion, it is one more independent piece of supporting evidence for the biological validity and general relevance of the findings which we describe here.

While we were able to show possible correlations across several pairs of the molecular levels that we assayed in this study, there are other combinations of molecular levels that show no obvious connection to each other in our data. We believe that this could be a function of either the developmental time point or the cell type, or both, being removed from those where the 22q11.2 and 1q21.1 deletions most likely exert some of their strongest effects (e.g., during embryonal development and in cells of the developing central nervous system). The clear effects on several genomic and epigenomic levels that we were able to observe in LCLs could represent the afterglow or reflection of a molecular tragedy that played out earlier in the development of various organs in the patients carrying these deletions. At the same time, the strength of this afterglow would hint at the strength of the effects that impacted across various levels of molecular control and gene regulation by CNVs of this size.

## Methods

**Cell lines**. All cell lines were either acquired from the Coriell cell repository (cell lines IDs starting with GM or ID) or were taken from the Molecular Genetics of Schizophrenia (MGS) cohort (dbGaP Study Accession: phs000167.v1.p, cell lines 52425 and 82699) and were appropriately consented.

**Hi-C libraries**. The Hi-C assay was carried out according to the original protocol[20], with several modifications. Twenty-five million cultured Epstein–Barr virus-transformed lymphoblastoid cells were spun down, crosslinked by adding 9 ml fresh medium and 0.25 ml 37% formaldehyde and incubated for 10 min at room temperature (RT). The reaction was stopped by adding 0.5 ml 2.5 M glycine and incubating for 5 min at RT. For cell lysis, 5 ml lysis buffer (500 µl 10 mM Tris-HCl pH8.0, 10 mM NaCl, 0.5% Ige cal CA630; 50 µl protease inhibitors (Sigma, St. Louis, MO)) was added to the cells followed by incubation on ice for at least 15 min, cells then were lysed with a Dounce homogenizer by moving the pestle A up and down 10 times, followed by incubating on ice for 1 min and 30–45 more strokes with pestle B. The resulting suspension was spun down, the supernatant was discarded and the pellet was washed twice with 500 µl ice-cold 1× NEBuffer 2 (NEB, Ipswich, MA). The pellet was then resuspended in 1× NEBuffer 2 in a total volume of 50 µl × n (n = number of tubes, at this point the sample can be split into multiple aliquots as needed). Next, 346 µl 1× NEBuffer 2 were added per tube. To remove proteins that were not crosslinked directly to the DNA, 1.9 µl of 20% sodium dodecyl sulfate was added per tube and the mixture was resuspended and incubated at 65 °C for 20 min, then at 37 °C for 30 min with shaking. Tubes were put on ice and 44 µl 10% Triton X-100 was added and mixed, followed by shaking for 30 min at 37 °C. Chromatin was subsequently digested overnight at 37 °C by adding 600 Units HindIII (NEB).

After digestion, the following several steps were carried out according to the original Hi-C publication[20]: marking of DNA ends and blunt-end ligation, DNA purification, removal of biotin from unligated ends, and shearing. After shearing, end repair, and "A" addition to the ends of DNA, 2 µl of Illumina adaptors (PE1.0 and PE2.0) were ligated to 1 µg of DNA using 2 µl T4 DNA ligase (Promega) and incubating at RT for 30 min. After purification with QIAquick spin columns, DNA fragments with adaptor attached were selected and enriched for biotin-labeling by biotin pull down using Streptavidin C1 beads. The resulting eluate of 50 µl DNA solution was divided into 20 individual PCR tubes with each containing 2.5 µl of the eluted DNA, 0.25 µl of Illumina paired end primers (PE1.0 and PE2.0), and 1× Phusion High Fidelity master mix with HF buffer (NEB) in a total volume of 25 µl. PCR was carried out with a temperature profile of 30 s at 98 °C followed by 9 cycles of 20 s at 98 °C, 30 s at 60 °C, 30 s at 72 °C, and a final 7-min extension at 72 °C. PCR products were pooled and purified by Qiagen MinElute column. DNA was then run on a 2% E-gel for the purpose of purification and size selection. DNA fragments between 450 and 600 base pairs were excised and purified with a gel extraction kit (Qiagen). DNA concentration was measured with the Quant-iT assay (Invitrogen). Finally, the Hi-C library was sequenced on the Illumina HiSeq platform using paired-end sequencing.

**Capture Hi-C libraries**. For capture Hi-C of chromosome 22q, we prepared separate in situ Hi-C libraries using the previously reported protocol[32]. Briefly, five million cells were crosslinked with formaldehyde and lysed with protease inhibitors (Sigma, P8340). Chromatins were subsequently digested by MboI restriction enzyme (NEB, R0147). Restriction fragment overhangs were filled and the DNA ends were marked with biotin. Next, biotinylated DNA were sheared to a size of 300–500 bp, whose ends were repaired subsequently. dATP were attached and

Illumina indexed adapter (TrueSeq nano) were ligated. The DNA fragments were then amplified and purified. A total of 2.1 million oligomer capture probes that densely tile the sequence of chromosome 22q were designed using the NimbleDesign Software and ordered from Roche Sequencing (formerly NimbleGen-Roche, Pleasanton, CA). Roche Sequencing's SeqCap EZ Choice XL Enrichment Kit was used to capture fragments from the in situ Hi-C libraries using the chromosome 22q-specific capture oligomers.

**ChIP-Seq libraries**. ChIP-Seq was performed according to the protocols in previous studies[44,45]. Briefly, per ChIP $2 \times 10^7$ LCLs were crosslinked in 1% formaldehyde for 10 min at RT. The reaction was stopped by adding glycine to a final concentration of 0.125 M and stirring for 5 min at RT. The cells were pelleted and washed with 1× phosphate-buffered saline (PBS) plus protease inhibitors to remove the crosslinking reagent. The cells were pelleted again and stored at −80 °C until further use.

On the day of the immunoprecipitation, the pellets were thawed and washed with 1× PBS plus Protease Inhibitors. The cells were pelleted and exposed to hypotonic buffer and broken with a Dounce homogenizer. The cell nuclei were pelleted and lysed in RIPA buffer. Chromatin was prepared using a Branson 250 Sonifier (7 × 30 s, 100% duty cycle). During the procedure, the lysate was kept cold at all times.

A sample of the chromatin lysate was set aside as input control. The rest of the chromatin lysate was used in the ChIP experiment using the following antibodies against H3K27Ac (Abcam ab4729, 1:110 dilution), H2K27me3 (Cell Signaling 9733, 1:175 dilution), and CTCF (Millipore 07-729, 1:175 dilution). Control IgG from the corresponding species was used as negative control. The Chromatin–antibody complexes were pulled down using Protein G Dynabeads (Life Technologies). The isolated DNA was purified and tested for successful enrichment using quantitative PCR against known loci. Illumina sequencing libraries were prepared using Illumina TruSeq adapters and the enzymes specified in previous study[45]. Libraries of the size range of 350–650 bp were excised from an agarose gel and cleaned up using the Qiagen Gel Extraction Kit. Libraries were PCR amplified for 14 cycles. Four to five libraries were pooled on one HiSeq2000 lane and sequenced using 2 × 100 bp.

**RNA-Seq libraries**. The RNA-Seq libraries were generated according as in a previous study[46]. Briefly, polyadenylated RNA fragments were purified using a Dynabeads mRNA Purification Kit, fragmented, and reverse transcribed into first-strand cDNA using random hexamer and Superscript II reverse transcriptase, followed by second-strand cDNA synthesis using RNaseH and DNA polymerase I. The resulting cDNA was end-repaired and a single "A" was added at the 3′ ends before ligating to Illumina paired-end sequencing adaptors. After running on an agarose gel, DNA fragments from 250 to 350 bp were cut out and extracted using a Qiagen MinElute Gel Purification Kit, and PCR amplified using Phusion High-Fidelity master mix and Illumina PE primers.

**Hi-C data analysis**. All Hi-C data were produced using Illumina paired-end sequencing with a read length of 2 × 101 bp. As there might be ligation junctions present in the reads, we performed iterative mapping using bowtie2 as in Imakaev et al.[23]. Briefly, we computationally cut all the reads to 25 bp first and mapped them to human genome (hg19). Then we extended the non-uniquely mapped reads by 5–30 bp and mapped them again. This process was repeated until the read length was extended to 101 bp. This iterative mapping did improve the mapping rate (Supplementary Table 5). Each read end was mapped separately using the single-end mode. Only uniquely mapping reads were used and PCR duplicate read pairs were removed. Self-ligation fragments and the read pairs whose sum of distances from mapped positions to the nearest restriction sites is larger than the length of the fragments in the Hi-C library were further removed by hicpipe[22]. We only included autosomes in our study. The filtered contact number is listed in Supplementary Table 6.

We compared three different data normalization methods: hiclib[23], hicpipe[22], and HiCNorm[24]. All of the three tools were run using the default parameters except for the segment length threshold being set to 600 bp. We chose normalized metrics on hicpipe for the following analyses, using a bin size of 40 kbp for topological domain analysis and of 500 kbp for the other analyses. The total number of contacts was normalized for each sample before combining cell lines in each category (control, 22q11del and 1q21del, respectively). Fold changes of log2-transformed mean contacts between deletion cell lines and control cell lines were calculated by (deletion−control)/control.

To identify the differential inter-chromosomal contacts, we only included contacts with at least one supporting read pair in each of the cell lines. Differential contacts analysis was conducted by two-sided Student's $t$ test with the "t.test" function in R using the normalized metrics. Fisher's exact test was used to assess the enrichment of differential contacts within the top 5% strongest contacts. We also performed the same analysis by permuting the control and 22q11del status of the cell lines ten times. Comparison within control cell lines and within 22q11del cell lines were performed by randomly dividing the cell lines into two groups three times.

**Generation of haplotype-specific chromosomal contact maps**. To phase the homologous chromosomes of two patient cell lines, we first carried out deep whole-genome sequencing (37.4× genome-wide coverage for ID00014 and 32.8× genome-wide coverage for ID00016), which yielded the required heterozygous SNV information. Haplotype phasing was then carried out by SLRH[21] in combination with Mendelian inheritance patterns of informative SNVs, based on the knowledge that ID00014 is a parent of ID00016. Briefly, genomic DNA was sheared to fragments of about 10 kbp length, which were diluted as needed and pipetted into a 384-well plate with 3000–6000 gDNA fragments in each well. Within each well, fragments were amplified, fragmented further, and ligated to barcodes unique to each well. Fragments from across all wells were pooled together and sequenced on Illumina HiSeq. Sequencing reads were mapped to the reference genome and assigned to their unique well based on their barcodes. For each well, reads were assembled into haplotype blocks based on their overlapping heterozygous SNVs. The haplotype blocks were then used to construct long haplotype contigs using Illumina's haplotyping algorithm Prism (v2.2). The N50 length of the haplotype contigs was 492,634 bp and 453,807 bp for ID00014 and ID00016, respectively. SNVs that could be phased by Mendelian inheritance analysis (heterozygous in one individual but homozygous in the other) were assigned to the phased haplotype contigs to assemble these haplotype contigs into whole phased chromosomes. The phased genome data of two control cell lines from related donors had been previously reported[21] and were included in the analysis. More details on this approach can be found in Supplementary Information.

Capture Hi-C reads were mapped by BWA-MEM[47] in single-end mode after ligation junction removal using Cutadapt[48]. Reads spanning the position of phased SNVs were assigned to the corresponding homologous chromosome using custom-written scripts. Then HOMER[27] was used to pair up the reads, filter the read pairs assigned to haplotypes (duplicates, self-ligations, read pairs <1.5× the sequencing fragment length, or distance to restriction site >1.5× the fragment length), and generate contact matrices for each homologous chromosome separately. To normalize the contacts involving the 22q11.2 region to the comparable level with contacts involving other regions on the intact homologous chromosome 22 in the patients, we calculated the expected number of phased heterozygous SNVs in this 3 million bp region, based on the density of phased heterozygous SNVs on chromosome 22q in each patient. Random single base pair positions in the 22q11.2 region were sampled to the calculated expected number. Only the contacts involving these sampled positions were included in the analysis. We performed this random sampling for ten times and the pattern remained consistently similar. We adjusted for recombination events for the haplotypes of ID00014 on the grounds that intra-chromosomal contacts should be more prevalent than inter-chromosomal contacts.

Identification of A and B compartments was performed as in Lieberman-Aiden et al.[20] and topological domains were identified as in Dixon et al.[33].

**Three-dimensional FISH**. Two human DNA BACs (clones RP11-47L18 and RP11-125K3) covering two distinct regions of chromosome 22 were labeled with biotin or digoxigenin (DIG) by the Nick Translation Kit (Roche Applied Sciences) to make FISH probes (Roche). In situ hybridization was performed according to the method published by a previous study[49], with several modifications. Briefly, GM12878 and GM06990 cells were immobilized on poly-L-lysine-coated slides for 1 h at 37 °C before being fixed with 4% paraformaldehyde and permeabilized with 0.05 % Triton X-100. Cells were hybridized with DIG-labeled and biotin-labeled FISH probes overnight at 37 °C. Hybridized samples were washed and then incubated with anti-DIG-Rhodamine (Roche, 1:100 dilution) and anti-biotin-Alexa488 (Invitrogen, 1:100 dilution) for 1 h at 37 °C. 3D two-color image stacks were captured by a Zeiss LSM510 confocal microscope. Distances between two FISH signals were measured by ImageJ.

**3D FISH data analysis**. For all individual cell lines tested in the FISH experiment, we first masked their identities to allow us picking target cells for high-resolution 3D imaging in an unbiased fashion. Between 16 and 33 cells were chosen randomly for each tested cell line, the basic condition for each cell being that 3 out of the 4 expected FISH signals had to be clearly visible. The distance between the targeted chromosomal regions was then determined by measuring the average distance between the green and red signals, normalized by the multiple of the shortest and longest radius of the nuclei.

To determine whether the distance between FISH signals is different between the 22q11 deletion and control groups, we carried out an analysis of variance with consideration of gender difference among the samples and distinct cell line subjects in each group:

$$\text{Distance} \sim \text{Deletion} + \text{Gender} + \text{Deletion/Subject}.$$

The parameter Subject is a random effect variable that represents the cell lines tested in our analysis. The variable Deletion indicates whether the cell line contains the 22q11 deletion and was initially masked when picking cells for imaging and signal distance measurement. The Subject parameter is nested within the Deletion parameter, as each cell line either has the 22q11 deletion or carries the full-length chromosome 22q.

**Differential expression analysis.** All RNA-seq data were generated using Illumina paired-end sequencing with read length 101 bp. Reads were mapped to hg19 and transcriptome reference with TopHat 2[50]. TopHat 2 was run using default parameters but with the coverage search being turned off. The mapped reads were analyzed by Cufflinks[51]. Differential expression was estimated with Cuffdiff 2[52]. We excluded the genes with low expression (FPKM < 0.5) from downstream analysis.

Pathway analysis of significantly differential expressed genes was conducted with DAVID[53] using all the expressed genes as background.

**Allele-specific gene expression analysis.** Samtools mpileup (v0.1.19) and BCFtools (v0.1.19)[54] were used to count the number of reads mapped from bam files of RNA-Seq data to each allele of the heterozygous SNVs. Binomial tests were performed to determine whether the percentage of RNA-Seq reads mapped to the alternative allele was significantly different from the mean frequency of the alternative allele of all heterozygous SNVs within each individual. Only heterozygous SNVs with read coverage >10 were included in the analysis.

**ChIP-Seq data analysis.** All ChIP-Seq data were generated using Illumina paired-end sequencing with read length 101 bp. Reads were aligned to hg19 with BWA-MEM using default parameters[47]. Reads with low mapping quality (<30) were removed. PCR duplicate reads were removed using Picard (http://broadinstitute.github.io/picard). As a quality control, we calculated the normalized strand cross-correlation coefficient (NSC) and relative strand correlation (RSC)[55] to assess the signal-to-noise ratios. All the data showed higher NSC than RSC (Supplementary Fig. 14). Replicates for the same cell line on average showed higher correlation than datasets from different cell lines (Supplementary Fig. 15).

For CTCF and H3K27ac, we used MACS2[56] to call narrow peaks with default parameters. For H3K27me3, we used the broad peak calling in MACS2. For all peak calling, we used the corresponding whole-cell extract input library as background. For differential bound analysis, we used the R package DiffBind[57] with the effective library size for read count normalization. Then DBA_DESEQ2 method was employed to conduct the differential bound analysis. Signal artifact blacklist regions were excluded from our analysis (http://hgdownload.cse.ucsc.edu/goldenPath/hg19/encodeDCC/wgEncodeMapability).

**Enrichment analysis.** For the enrichment analysis of differentially expressed genes, we divided each chromosome into 500-kbp bins. Within each bin, we calculated the total number of expressed genes and the number of genes with significantly differential expression between 22q11del and control cell lines. Then we conducted Fisher's exact test to identify bins enriched with significantly differentially expressed genes against the background of the whole genome.

For the analysis of differentially enriched sites for CTCF, H3K27ac, and H3K27me3, we also used 500-kbp bins. Log2-transformed fold changes of normalized read numbers in binding sites between 22q11del and control cell lines were further transformed to Z-scores. We considered binding sites with Z-score >2 or <−2 as significantly bound sites. Then, within each 500-kbp bin, we calculated the total number of binding sites and the number of significantly differentially enriched sites between 22q11del and control cell lines. Then we conducted Fisher's exact test to identify bins with significantly differentially enriched sites against the background of the whole genome.

**Correlation between gene expression and histone modification.** To estimate the correlation between gene expression and histone modification, we assigned each binding site of H3K27ac and H3K27me3 to its nearest ENSEMBL TSS using the R package ChIPpeakAnno[58]. If a TSS was associated with multiple binding sites, only the nearest binding site was retained. We then used this assignment for downstream correlation analysis. To determine the cutoff for the distance in which binding sites are associated with TSSs, we plotted the distribution of distances between binding sites and their assigned TSSs (Supplementary Fig. 16). Based on the distribution, we set the cutoff to distance to TSS to ±1 kbp for H3K27ac and to ±5 kbp for H3K27me3.

To investigate the effects of histone modifications on gene expression, we divided the genes into two categories based on the differential expression analysis: differentially expressed genes (FDR < 0.05) (category A) and not differentially expressed genes (FDR > 0.05) (category B). Within each category, we further categorized the genes into two groups: upregulated expressed genes (A1, B1) and downregulated expressed genes (A2, B2) in 22q11del cell lines relative to control cell lines. Then, for the genes within each of the four groups (A1, B1, A2, B2), we calculated the Z-score transformed fold changes of the normalized read counts in the TSS-binding sites of histone marks between 22q11del and control cell lines. To obtain the statistical significance of the fold change differences between A1 and B1 and between A2 and B2, we performed permutation tests with 9999 permutations.

We also carried out the reverse analysis. TSS-binding sites of histone marks were partitioned into two categories based on the differential bound analysis: differentially bound sites (|Z-score| > 1 for H3K27ac, |Z-score| > 2 for H3K27me3) (category A) and non-differentially bound sites (|Z-score| < 1 for H3K27ac, |Z-score| < 2 for H3K27me3) (category B). Within each category, we further categorized the binding sites into two groups: upregulated bound sites (A1, B1) and downregulated bound sites (A2, B2) in 22q11del cell lines. Then we calculated the fold changes of the genes' FPKM between 22q11del and control cell lines within

each group. Permutation tests were performed with 9999 permutations to obtain statistical significance.

To estimate the correlation between gene expression and histone modification in a direct way, we calculated Pearson's correlation coefficient between gene's FPKM and normalized read counts in the corresponding TSS-binding site for each gene across all of the cell lines. To obtain statistical significance, we first permuted genes' FPKM across the cell lines for each TSS ten times to assess the background correlation levels and then performed the Wilcoxon rank sum test between the observed correlation coefficients and the background correlation coefficients. We also performed the same analysis using the differentially expressed genes only and differentially enriched binding sites only.

**Correlation analysis for epigenetic marks.** To assess the correlation of epigenetic marks binding between different regions on the same chromosomes, we divided the chromosomes into 500-kbp bins. Within each bin, we calculated the mean value of the normalized read counts for all the binding sites of each epigenetic mark. Then we calculated the Pearson's correlation coefficients of obtained mean values across the cell lines between any two bins on the same chromosomes.

## Data availability

Hi-C, ChIP-Seq, and RNA-Seq data from this study have been submitted to the NCBI Gene Expression Omnibus (GEO; http://www.ncbi.nlm.nih.gov/geo/) under accession number GSE76922. All other relevant data supporting the key findings of this study are available within the Article and its Supplementary Information files or from the corresponding author upon reasonable request.

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

## Acknowledgements

This research was supported by external funding, to A.E.U., from the National Institute of Mental Health (NIMH) (grant number: MH100010), the National Human Genome Research Institute (NHGRI) (grant number: HG007735-01, Center PI Howard Chang, Project PI A.E.U.), NARSAD (grant number: YI Award 19673), and the March of Dimes (grant number: #6-FY13-142), as well as by Stanford University funds to A.E.U., from the Stanford Pediatric Research Fund (grant number: UL1 RR025744) and from the Stanford Department of Psychiatry and Behavioral Sciences and the Stanford Department of Genetics. A.E.U. is a Tashia and John Morgridge Faculty Scholar of the Stanford Child Health Research Institute. We thank Dr. Eitan Yaffe (Stanford University) for his advice on the use of hicpipe.

## Author contributions

X. Zhang contributed to study design, coordinated data production and carried out data analysis, and contributed to writing the paper; Y.Z. contributed to study design, produced experimental data and contributed to data interpretation; X. Zhu contributed to study design and carried out data analysis; C.P. produced experimental data and contributed to data interpretation; M.S.H., T.W., A.K., and J.Y. produced experimental data, S.M.W. contributed to study design, contributed to directing data production and contributed to data interpretation; A.E.U. conceived of and designed the study, directed data production, data analysis and data interpretation, and wrote the paper.

## Additional information

**Competing interests:** The authors declare no competing interests.

