## [peer review file · Nature Communications]

Reviewer #1 (Remarks to the Author):

This is a well-written article describing the impact of a large recurrent CNV associated with neurodevelopmental disturbances and congenital malformations on chromosomal organization and interactions, epigenetic modifications and expression patterns. The authors used state of the art techniques (described in sufficient detail) to address this relevant question and obtained novel findings that are important to the field. Namely, they observe changes in intra- and inter-chromosomal contacts involving the deleted regions, but also more global changes in chromosomal organization. Interestingly chromatin modifications were seen to be altered not only within but also in regions flanking the CNV (up to 2Mb away), with direct impact on gene expression levels. Although there is at least another report of this kind for another neuropsychiatric disease (NPD) related CNV, this is one the first studies in this theme and the use of high-throughput and unbiased approaches makes it very interesting. Validation of the key findings for yet another NPD strengthens the conclusions.

One important limitation of this work is that the analysis is performed in lymphoblastoid cell lines. For one, this is not the most relevant cell type for the CNV in question, but equally important is the fact that during the transformation process LCLs often acquire additional chromosomal rearrangements, which may have an impact on the results. This should be excluded in the cell lines used in this study, by aCGH or NGS methodology.

One intriguing aspect of several CNVs associated with neurodevelopmental disorders is the incomplete penetrance and variable expressivity they are associated with. While in this study the authors explicitly focused on the aspects shared by all patients, it would also be interesting to study the associations of specific changes in the chromosomal landscape with specific clinical features. Perhaps they could at least comment on this possibility in the Discussion.

Specific comments:

Page 7. authors mention a "dosage effect" however they only have the control vs. hemizygous state for the 22q11.2 region, not homozygous deletions or duplications of the region, therefore this term seems inadequate.

Page 10, lines 194/195: what is the largest other affected region?

Page 10, line 211: do the authors propose any explanation for the differential binding of CTCF only in the region downstream of the deletion?

In Figure 2, immediately down stream of the deletion there seems to be a region where all modifications and CTCF binding are increased. is this so? is there any explanation for this?

Reviewer #2 (Remarks to the Author):

In this study, the hypothesis is tested that the 22q11 recurrent deletion not only affects genes in the region but affects also the broader nuclear architecture and hence, has a genome wide effect on gene expression. To test this hypothesis, Hi-C analysis as well as ChIP-Seq and transcriptome analyses were performed. Not surprisingly, they see effects on the 3Mb deleted region. In addition, the data suggest genome wide perturbations on nuclear architecture and gene expression. To further support this claim, another recurrent microdeletion 1q21.1 is investigated. This is a large body of descriptive work with interesting and important observations. These observations are interesting since they may change the current paradigm that the deletions are directly affecting the phenotype to a novel paradigm that large CNVs effects may multilayered and indirect.

Major comments:

1. Compartmentalisation: the authors claim that the original compartment that is spanning the deletion region in control cells is being partitioned into two shorter compartments. I have two questions, a philosophical and a technical: (a) The cell lines contain a normal and a deleted chromosome 22. There is no reason why the topological domains in the normal chromosome 22 would change. The chromosome carrying the deletion cannot have interactions within the deleted region. The data presented, both in the 22q11Deletion and normal chromosome are a mix of interactions on both chromosomes. Hence, the reconstituted domains are an artificial combination and not real. I wonder whether you can keep this claim. (b) on a technical note, although the authors draw a different topological domain, looking at the data suggest that the original domains are also there. What is the strength of the data to claim to different domains?
2. Page 16: the 3D topology is measured by 3D FISH experiments. In the supplementary figure 6, the distances are measured between the telomeric and the deletion proximal region and a difference is observed between cases and controls. There should also be a difference between the normal and deleted chromosome within the 22q11DS cases. This difference should be more outspoken. I suggest to show/plot those data.
3. The data on 1q21.1 presented are limited. It would be valuable to also show the genome wide contact changes.

Minor comments:

- Abstract: Conclusions: 'These findings suggest novel principles': Describe/Name those novel principles.

- The introduction is not very focused and suggest limited knowledge of clinical genetics: Why is there a focus on the difference between small /medium CNVs and large CNVs. This is not relevant for the rest of the article. Line 67: schizophrenia, ASD are placed at the same level as Williams Syndrome, a well delineated developmental syndrome. Line 72: The authors suggest that neuropsychiatric CNVs have been mostly studied by applying the paradigm of trying to determine wich single gene is important. In the field, both such CNVs are termed contiguous gene syndromes, arising from the observation that multiple genes concomitantly cause the phenotype. Rephrase. Line 78: add references. Line 97; 98: Add references (f.e. look at/within McDonald McGuinn, Nature Primer, 2015).
- It is mentioned that other studies showed an effect on gene expression outside the CNV. Refer to those studies (f.e. Harewood et al. Methods Mol.Biol.2012)
- Results: Line 125-128: Replace this paragraph to later?
- Throughout the text: deletion-downstream: better to use the term deletion-distal and deletion-proximal.

Reviewer #3 (Remarks to the Author):

In the manuscript “Local and global chromatin interactions are altered by large genomic deletions associated with human brain development“ Zhang et al. describe a Hi-C, ChIP-seq, and RNA-seq analysis of immortalized lymphoblastoid cell lines from 5 patients carrying a 22q11.2 deletion in order to study the local and global effects of such deletions on chromatin folding, epigenetic modifications and gene expression. They find that the circa 3Mb deletions mostly affect gene expression and chromatin folding at the 22q11.2 region of the genome and do not find systematic global effects on chromatin organisation. Interestingly, they show that the 3Mb deletions seem to decrease the spatial distance between the 22q11.2 region and a telomeric region of chromosome 22.

The experimental approach and generated dataset are very relevant from a medical genetics viewpoint and are promising to offer important insights into the disease etiology and molecular pathogenesis of CNVs in general and for 22q11.2 deletions in particular. However, the current amount and quality of the analysis is only a starting point for such a paper and mostly confirms previously published results at other loci (Mundlos lab for congenital disease, Korbelt lab for cancer) without providing substantial advance in our understanding of chromatin folding and disease. Aside from presenting their data in a sometimes rather difficult to follow way, what is it that we learn from this study?

The authors touch at least two interesting points that could reveal interesting biology if followed up in more detail; the switch of A/B compartments and the interaction between very distal parts of chr22. For the A/B switch: How reproducible is this? Is this seen in every patient cell line and every control cell line? Does this correlate with called regions of differentially regulated genes and chromatin marks? The very long range interaction. Is there a large structure of chromosome 22 in which centromeres and telomeres of the chromosome are in close chromosomal contact? The deletion would bring RP11-47L18 very close to the centromere. Or what is the molecular relevance of this chromosome-scale observation? Does this affect gene expression?

Aside from this, there are additional issues where the data analysis seems incomplete:

- The authors importantly state the problem of normalization in heterozygous deletion alleles and overcome this problem by using the hicpipe algorithm. For a fully informative analysis of the data, the interaction on the allele carrying the deletion should be at least be approached, e.g. by subtracting half the wildtype signal from the mutant, thereby providing a view of the “mutant allele only”. I agree that haplotype-phasing would be the optimal, but very difficult thing to do and would warrant a publication in its own right.
- Hi-C data: Similarities and differences between deletion cell lines (Figure 1 and 3). Have the authors always combined the Hi-C data from 5 patient cell lines to perform their analysis? How do the alleles differ and why do they need to/why should they be combined in the analysis? How reproducible between individuals are detected contact and gene expression changes. As shown in Figure 1a the deletion creates contacts between the now adjacent parts of the genome. The findings from the Mundlos lab would predict that these contacts extend only to the end of the next TAD or some other similar structure. Is this the case? Do we get a new, fused TAD between the upstream and downstream regions of the deletion? The resolution in Fig. 1a is 500kb and cannot be used for this and the interaction map in Figure 3b actually cuts out this highly informative piece of information and from what is presented, the data looks very sparse. Also genes, gene expression and ChIP-seq tracks should be included in such a figure.
- If the sequencing depth is not sufficient to create maps of higher resolution (which might possible, see Sup Tab. 5) it might make sense to increase sequencing depth for just one or two individual experiments to create a high-res maps, that allow a better view of the TADs than what is currently presented.
- Chromatin marks and gene expression: The authors find that H3K27ac and gene expression in the 22q11.2 flanking regions are affected. How do the decreased H3K27ac and increased H3K27me3 marks correspond to the downregulated BID and IGLL5 genes? Are these individual

enhancers, only the promoter of the gene etc.? Only showing the 500kb binned data is not sufficient. Moreover, it should be possible to use the SNP information in Hi-C and RNA-seq data to assess the allele-specific changes in gene expression (i.e. expression from the “wildtype” allele should be unaffected, the 22q11 del-allele should have weaker expression?)

- Analysis of ChIP-seq data (Figure 2): With ChIP-seq the authors have potentially enhancer-level information on histone modifications (see also above). However, they decide to analyse it by plotting the p-value of differentially enriched marks in 500kb windows, not giving information how they relate to wildtype or newly formed TADs, compartments or differentially expressed genes. What are we supposed to learn from these graphs?

In the below please find our detailed answers to the reviewer's comments, and descriptions of the nature of the revisions and additions to our manuscript. We thank the reviewers for their thoughtful and detailed critiques and comments. We feel that the resulting revisions strengthened the manuscript considerably.

Reviewer 1

Reviewer #1 (Remarks to the Author):

This is a well-written article describing the impact of a large recurrent CNV associated with neurodevelopmental disturbances and congenital malformations on chromosomal organization and interactions, epigenetic modifications and expression patterns. The authors used state of the art techniques (described in sufficient detail) to address this relevant question and obtained novel findings that are important to the field. Namely, they observe changes in intra- and inter-chromosomal contacts involving the deleted regions, but also more global changes in chromosomal organization. Interestingly chromatin modifications were seen to be altered not only within but also in regions flanking the CNV (up to 2Mb away), with direct impact on gene expression levels. Although there is at least another report of this kind for another neuropsychiatric disease (NPD) related CNV, this is one the first studies in this theme and the use of high-throughput and unbiased approaches makes it very interesting. Validation of the key findings for yet another NPD strengthens the conclusions.

We are very thankful for these comments. These were the aims we had with our study and it is very encouraging to read that in general we did achieve them. It is our opinion as well that these kinds of analyses are justifiably done for many disease associated CNVs, and certainly for each of the 'major' CNVs, separately. Both differences as well as commonalities that result from such work will be of importance.

1. One important limitation of this work is that the analysis is performed in lymphoblastoid cell lines. For one, this is not the most relevant cell type for the CNV in question, but equally important is the fact that during the transformation process LCLs often acquire additional chromosomal rearrangements, which may have an impact on the results. This should be excluded in the cell lines used in this study, by aCGH or NGS methodology.

We entirely agree. One major reason to choose LCLs was the desire to make it possible to carry out the various genomics assays that typically require millions of cells to produce robust results, or even to work at all (i.e. Hi-C, ChIP-Seq). Future years will bring technological advances that will make it possible to replicate our findings in cell types such as neurons derived from iPSCs from 22q11DS patients. We think that it will be work using LCLs that has opened the door and pointed into the right directions. Stem cell based analyses will build on and reference the LCL-based work. So we feel that it was a useful and necessary undertaking to add to the foundation of this area of research while using LCLs. This point is underlined by the

only other work so far that also examined the effects of a different large neurodevelopmental CNV also having used LCLs (the 16p11 CNV, Loviglio et al., *Molecular Psychiatry* 2016 – using 4C instead of Hi-C, no haplotype specific analyses, a smaller number of cell lines and no CHIP-Seq data from the same patient cell lines).

We realize that it is important to be exceedingly restrained with statements regarding the various clinical phenotypes in 22q11DS based on LCL work, and we have been careful to do so in this manuscript. Rather, our aim is to draw conclusions of a basic and general nature on the molecular level about the impact of large CNVs in general, and the 22q11.2 CNV in particular.

Lastly, it cannot be ruled out that some genes that are part of changes in molecular networks are of importance in multiple cell and tissue types. But we were very careful not to present any far-reaching conclusions in this direction. We used the LCLs as a point of entry into this important area of research.

Regarding the second point, the potential genomic instability of LCLs in tissue culture: we now have included analyses to control for this. We have carried out whole-genome sequencing analysis for the patient lines at a genomic sequence coverage comparable to that used by the 1000 Genomes Project (i.e. <10x genome wide coverage). Almost all of our control cell lines had been included in the 1000 Genomes Project and therefore we can now compare the genome-wide CNV load between our patient LCLs and the control LCLs. The result of this analysis is that there is no elevated genome-wide CNV burden in our LCLs (Supplementary Table 2). We had already previously confirmed the presence and boundaries of the 22q11.2 deletion in our patient LCLs using whole-genome sequencing (Supplementary Figure 1).

2. One intriguing aspect of several CNVs associated with neurodevelopmental disorders is the incomplete penetrance and variable expressivity they are associated with. While in this study the authors explicitly focused on the aspects shared by all patients, it would also be interesting to study the associations of specific changes in the chromosomal landscape with specific clinical features. Perhaps they could at least comment on this possibility in the Discussion.

This is a very important point. The question of phenotypic variance is always on our minds when studying 22q11DS. But we felt that with the cohort size and cell type available for this study we should not draw any conclusions in this regard. We have added the following paragraph (lines 567-573 in the revised manuscript or lines 670-676 in the changes tracked version) to the Discussion:

Our study does not address the question of what the molecular causes for phenotypic variance between individual carriers of the 22q11.2 deletion might be. This is a highly important question but one that, given the vast number in each individual genome of potential genetic modifiers that together with environmental factors are the prime suspects for such variance, will need far larger cohort sizes of cell lines modeling the tissues where such variance is observed. Such cohorts are currently being assembled and will be the basis of highly interesting work in the coming years.

Specific comments:

3. Page 7. authors mention a "dosage effect" however they only have the control vs. hemizygous state for the 22q11.2 region, not homozygous deletions or duplications of the region, therefore this term seems inadequate.

Thanks for pointing that out. We have changed the manuscripts to use the terms 'decrease' and 'increase' instead, when referring to changes in chromosomal interactions.

4. Page 10, lines 194/195: what is the largest other affected region?

This is a very good question. We searched across the whole genome in the ChIP-Seq data and found only four other regions with decreased H3K27ac, H3K27me3 and CTCF binding signals in patient cell lines, i.e. where there is consistent ChIP-Seq signal change into one direction across the entire length of the genomic segment. The largest of these four regions is 502 kbp in length, on chromosome 1, as shown below (from top to bottom: H3K27ac, H3K27me3, CTCF; dashed lines mark the boundaries of the region):

However, the decrease of the ChIP-Seq signals in this region was not as dramatic as for the 22q11.2 region. The lengths of the other three regions were 120 kbp, 150 kbp, and 375 kbp respectively (on chromosomes 3, 19, and 7, respectively), i.e. much smaller than the 22q11.2 deletion region, and the fold changes in ChIP-Seq signal were small as well. No genes in these four regions were differentially expressed between patient and control cell lines.

We thank the reviewer for this question. Now having checked for this confirms that the 22q11.2 deletion stands out from the rest of the genome regarding uniform changes in ChIP-Seq signal over an extended length of genomic sequence.

5. Page 10, line 211: do the authors propose any explanation for the differential binding of CTCF only in the region downstream of the deletion? In Figure 2, immediately down stream of the deletion there seems to be a region where all modifications and CTCF binding are increased. is this so? is there any explanation for this?

We have been wondering about these questions in the lab. These are some of the central findings of the paper. We can speculate about explanations and we think these findings will be stimulating important discussions in the field – and where extensive follow-up studies will be needed to reach conclusive answers on the molecular level.

Histone marks are differentially changed in both the deletion-upstream and deletion-downstream regions. Meanwhile CTCF binding is also differentially changed, but only in the deletion-downstream region.

One could speculate that since the deletion can be expected to have occurred before the changes in chromatin marks and protein binding (i.e. at some point during meiosis) that it was the change in chromosome interaction patterns forced by the physical shortening of the chromosome that then lead to the changes in chromatin marks. Regarding changed CTCF binding, there is one more striking difference between the deletion-upstream and deletion-downstream regions, namely that the deletion-downstream region is also showing increased intra-chromosomal interaction with the region at the very telomeric end of 22q. So it could be that this deletion-downstream-telomeric interaction is involved in the changes in CTCF binding while the histone mark changes on both sides of the deletion are mainly driven by the increase in deletion-proximal chromosome interactions. It would remain to be seen whether the CTCF-binding changes precede the changes in deletion-proximal-telomeric interactions or are a result of those interaction changes. Also open is whether all change in ChIP-Seq signal are mediated purely via chromosome interaction changes or whether there are more indirect mechanisms resulting from the widespread gene-expression network effects caused by the deletion.

However, we did not want to include these speculations in the manuscript due to lack of experimental evidence. We think that this is one of the things that will

have to be discussed by the field and then followed up now that we have those findings.

The region downstream of the deletion (23-23.5 Mbp) was indeed enriched with changes in histone modifications and CTCF binding. However we would like to clarify that in the original Fig. 2 (Fig. 3a-c in the revised version), the $-\log p$ values (shown as y axis) were for the enrichment analysis, which was used to identify which regions were significantly enriched with differential binding of histone modifications and CTCF between patients and controls compared to the whole genome background. In other words, histone methylation marks and CTCF binding are up while histone acetylation is down. What Fig. 3a-c shows is that differential ChIP-Seq signals reach significance for this region without specifying the direction of the change.

For instance, 11 out of 30 H3K27me3 marks in this region showed significantly differential signal between patients and controls compared with 3,031 out of 59,912 H3K27me3 marks genome-wide. Fisher's exact test of enrichment of differential signals in the 23-23.5 Mbp region showed this differential signal to be extremely significant (p-value of 1.26E-07). All of the 11 differentially changed signals for H3K27me3 showed stronger signal in the patient cell lines. These results, together with the signals for H3K27ac and CTCF marks, were described in the third and fourth paragraph in the Section "Chromosome contacts increase across the 22q11DS breakpoint junction and chromatin marks in its flanking regions are affected in a concerted manner" of the Results. As region 23-23.5 Mbp was close to the 22q11.2 deletion, its interactions with the upstream region of the deletion increased in the patients. We speculate that both chromosome conformation changes and gene expression network changes might contribute to the change of histone modifications and CTCF binding.

Reviewer 2.

Reviewer #2 (Remarks to the Author):

In this study, the hypothesis is tested that the 22q11 recurrent deletion not only affects genes in the region but affects also the broader nuclear architecture and hence, has a genome wide effect on gene expression. To test this hypothesis, Hi-C analysis as well as ChIP-Seq and transcriptome analyses were performed. Not surprisingly, they see effects on the 3Mb deleted region. In addition, the data suggest genome wide perturbations on nuclear architecture and gene expression. To further support this claim, another recurrent microdeletion 1q21.1 is investigated. This is a large body of descriptive work with interesting and important observations. These observations are interesting since they may change the current paradigm that the deletions are directly affecting the phenotype to a novel paradigm that large CNVs effects may multilayered and indirect.

We very much agree with the reviewer's interpretation of the motivations for and results of our study.

Major comments:

1. Compartmentalisation: the authors claim that the original compartment that is spanning the deletion region in control cells is being partitioned into two shorter compartments. I have two questions, a philosophical and a technical: The cell lines contain a normal and a deleted chromosome 22. There is no reason why the topological domains in the normal chromosome 22 would change. The chromosome carrying the deletion cannot have interactions within the deleted region. The data presented, both in the 22q11Deletion and normal chromosome are a mix of interactions on both chromosomes. Hence, the reconstituted domains are an artificial combination and not real. I wonder whether you can keep this claim. (b) on a technical note, although the authors draw a different topological domain, looking at the data suggest that the original domains are also there. What is the strength of the data to claim to different domains?

(a) We agree wholeheartedly. We were aware of this issue and had thought that we would have to leave it unresolved, originally simply reporting that the topological domain signal on top of the deletion in the patient cell lines is being partitioned into two shorter topological domains. This was a true statement that pointed to necessary follow-up analysis, but we always felt that it was nevertheless an unsatisfying situation as it left unanswered one of the most important questions, of what is happening on the two chromosomes 22q separately. We had already considered technical options for how to answer this question and then, spurred by the comments of the reviewers, decided to try and solve this problem using a novel approach of combining genome-phasing using virtual long reads and Mendelian genotype patterns with very-deep Hi-C using chromosome-wide capture for the enrichment of informative Hi-C fragments.

We enjoyed the challenge of establishing this approach and think it was well worthwhile to do so. We think that in addition to allowing a much better understanding of the effects of the 22q11.2 deletion this approach is also a technical achievement that will inform future work by others in the field of chromosome interaction analyses.

The haplotype-specific analyses revealed that the topological domains on the intact chromosome 22q remain unchanged while on the chromosome 22q with deletion the topological domain that was spanning the deletion is entirely absent while the domain on the upstream side of the deletion increases its interactions not just with the domain directly downstream but also with the two domains immediately following in the downstream direction.

We had mentioned in the original Discussion (lines 543-551) that haplotype phasing of the genomes, to be done in future studies, would be needed to distinguish the intact chromosome 22 and the chromosome 22 with the deletion.

Inspired by this question, which was echoed by reviewer 3 in comment #4, we decided to perform haplotype phasing on two patient cell lines from related donors (ID00014 is a parent of ID00016) to distinguish the Hi-C signal coming from the two

homologous chromosomes. Details of the haplotype phasing have been added to the Methods as well as to the Supplementary Information and we are describing the approach that we developed for this task in the passage below.

The method we used for haplotype phasing was that of statistically aided, long-read haplotyping (SLRH) (Kuleshov et al. *Nat. Biotechnology*. 2014). Briefly, deep whole genome sequencing was performed to obtain the single nucleotide variants (SNVs). Then genomic DNA was sheared into fragments of about 10 kbp, diluted and distributed into a 384-well plate. Each well contains 3,000–6,000 molecules, calculated to assure that a given locus will almost never be found more than just once in a well. The fragments within each well can then confidently be assigned to their proper haplotypes after sequencing. Within each well, fragments were amplified, cut into short fragments and barcoded with a unique barcode for each well. The fragments from all wells were pooled together and sequenced.

Sequencing reads were mapped to the reference genome and sorted to their original well based on the barcode specific to each well. Within each well, reads were assembled into haplotype blocks using their overlapping heterozygous SNVs, which were then assigned to long haplotype contigs based on a phased reference panel using the haplotyping algorithm Prism developed by Illumina.

To assemble these haplotype contigs into whole chromosomes, SNVs which can be phased based on Mendelian inheritance considerations (i.e. heterozygous in one of the two patients but homozygous in the other) were assigned to these phased haplotype contigs. For instance, if the genotype of a SNV in ID0016 is A/C but A/A in ID0014, we were able to infer that allele A in ID0016 was inherited from ID0014. The whole haplotype contig of ID0016 containing allele A at this SNV position was from ID0014 too. After we connected all the haplotype contigs inherited from ID0014 we obtained the whole chromosome haplotype in ID0016 inherited from ID0014. Connecting the haplotype contigs that are not inherited from ID0014 generated the other whole chromosome haplotype of ID0016. The total number of autosomal heterozygous SNVs phased on the whole chromosome level was 1,868,316 and 1,870,948 for ID0014 and ID0016 respectively.

For phasing the control lines we used the SLRH data for GM12878 and GM12892 that was already available (Kuleshov et al. *Nat. Biotechnology*. 2014). In combination with Mendelian inheritance patterns we obtained 1,929,967 and 1,874,181 autosomal heterozygous SNVs phased on the chromosome level for GM12892 and GM12878, respectively, which was comparable with the results for the patient genomes.

With the haplotype phasing data of two patients and two controls in hand we were able to distinguish the chromosomal contacts of the chromosome 22q with deletion from those of the intact chromosome 22q, in the patient cell lines, and also analyze the chromosome contacts separately for the two chromosomes 22q in the control cell lines.

However, we found that the Hi-C data we already had were not deep enough for obtaining high-resolution haplotype-specific contacts. We thus decided to perform capture Hi-C of chromosome 22q on these four cell lines with haplotype phasing data to obtain high coverage of chromosome 22. We prepared new Hi-C libraries using in situ Hi-C (Rao et al. Cell. 2014). Tiled and dense DNA probes were designed with NimbleDesign Software for the sequence of chromosome 22q. Roche NimbleGen's SeqCap EZ Choice XL Enrichment Kit was used to capture for human chromosome 22q in the in situ Hi-C libraries. With this capture Hi-C data we achieved from 10- to 16-fold enrichment of chromosomal contacts involving chromosome 22q compared with non-capture Hi-C data (Supplementary Fig. 3).

Based on the haplotype specific analyses, we removed the original statements regarding the A/B compartments and topological domains as well as the original Fig. 3 in the text. Instead we now show the haplotype specific A/B compartments and topological domains in Fig. 3d, Fig. 4 and Supplementary Fig. 6-7 using our newly generated haplotype phasing and capture Hi-C data on two patients and two controls.

(b) This is a very good question. We agree that even though the topological domain calling algorithm called two domains over the 22q11.2 region in the deletion cell lines, it does not necessarily mean our data is strong enough to support this claim as the algorithm might not be able to reflect the real domains under all circumstances. We have now deleted this passage to avoid overstatement and by ways of agreeing with the reviewer's comments that the originally reported domains were the result of a mixture of signals coming from the two different chromosomes 22q. The original Fig. 3 was also removed. We show the haplotype-specific A/B compartments and topological domains instead.

2. Page 16: the 3D topology is measured by 3D FISH experiments. In the supplementary figure 6, the distances are measured between the telomeric and the deletion proximal region and a difference is observed between cases and controls. There should also be a difference between the normal and deleted chromosome within the 22q11DS cases. This difference should be more outspoken. I suggest to show/plot those data.

If we understand this comment correctly we think this is something that cannot be further resolved with the 3D FISH approach. We understand the question to be whether in the patient cell lines it is always the chromosome 22q with deletion that shows a closer proximity between the FISH probes, relative to the intact chromosome 22q. With 3D FISH we cannot distinguish which chromosome 22q in a given microscopy image is the chromosome with deletion.

Of course the assumption would be that in most cases the chromosome where the two probes are closer together is the one with deletion. But the folding that brings the telomeric end of 22q and the deletion proximal region closer together is a dynamic process. We think that the Hi-C signal that indicates an increase in interactions between these two regions should be interpreted as this interaction happening more frequently and/or lasting longer for the chromosome 22q with

deletion. But it will also happen at some rate for the chromosome 22q without deletion. Therefore while it is much more likely that the two probes in closer proximity are on the chromosome 22q, in a given individual FISH image this cannot be determined with certainty.

To illustrate this point in a different way: in Figure 5 (original Figure 4) in the upper panels showing the FISH signal for a control cell one can see that even in a control cell the red and green signals are not at exactly the same distance from each other for the two (intact) chromosomes 22q. The two chromosomal regions in question will also interact with each other on intact chromosomes 22q, just less intensely, and the FISH image is a snapshot during that interaction.

It is when measuring over large numbers of control and patient cells across the probands that it becomes apparent that overall the distance between red and green probes are significantly shortened in the patient cells, as would be expected from the Hi-C signal.

This is why we carried out the statistics analysis for 3D FISH as reported. We determined the distances between the red and green signals over a large number of both individual control and individual patient cells across all of the cell lines used for Hi-C. The resulting measurements then result in the statistics reported in Supplementary Figure 9.

On the other hand the newly available haplotype specific Hi-C data should in principle allow determining the distribution of interaction intensity between intact and deletion-carrying chromosome 22q. We looked into our newly generated capture Hi-C data in the four haplotype phased cell lines to check if the contact between the downstream deletion-distal region and the telomeric region was different between the intact and deleted chromosome 22q in the patient cell lines. We did indeed see increased contact for those regions on the chromosome 22q with deletion relative to the intact one within the patient cell lines. We also observed small differences of contact between the two homologous chromosomes within the control cell lines, although these differences were smaller than the ones observed within the patient cell lines (underlining the point made above about the telomeric/deletion-distal contacts resulting from a process that is present for both chromosomes but increased for the 22q with deletion). However, for the haplotype specific Hi-C data we felt that we could not make such a determination in a statistically meaningful way as the signal is only from 2+2 cell lines (following the same reasoning that also led us to not draw too many conclusions for the two cell lines with 1q21 deletion for which we had Hi-C data). The finding is still significant when integrating the Hi-C signal coming from all five cell lines (and then was validated by 3D FISH).

To illustrate this point we are showing below the contact heatmaps of haplotype 1 (chromosome 22q with deletion for the patients) minus haplotype 2 (intact chromosome 22q for the patients) for each phased cell line (green is negative values; red is positive values; left panel is patients and right panel is controls):

The differences for the telomeric/deletion-distal interaction between the patient and control cell lines are visible. But, again, with data from only 2+2 cell lines this increased telomeric/deletion-distal interaction cannot be demonstrated quantitatively in a statistically sound fashion, as opposed to what was possible with the Hi-C data (and the 3D FISH data) from all the cell lines combined. We of course prefer caution to overstatement and therefore did not include these haplotype-specific data as evidence for increased contact only for the chromosome 22q with deletion, even though this is certainly by far the most probable explanation.

3. The data on 1q21.1 presented are limited. It would be valuable to also show the genome wide contact changes.

Thank you for this suggestion. We have now added the genome-wide contact changes for 1q21.1 as Supplementary Fig. 12. Since we only had Hi-C data for two cell lines with 1q21.1 deletions, the statistical tests for this analysis are not as robust as for the analogous analysis for the 22q11.2 deletion lines. This was the reason why we originally only intended to use the 1q21.1 data to show that the contacts between the flanking regions of the 1q21.1 deletion increased and all other contacts involving the 1q21.1 deletion region decreased in the cell lines with deletions compared with control cell lines. However, having Figure 12 in the Supplement is indeed an interesting additional item for this study.

Minor comments:

4. - *Abstract: Conclusions: 'These findings suggest novel principles': Describe/Name those novel principles.*

We had very early on expanded more on this point but there was a 250-word limit for the Abstract and we are currently already up against that limit. As it is there is now a new 150-word limit for Abstracts at Nature Communications. We would have to learn whether the editors would allow us to stay with a 250-word Abstract or not and then based on that we can rephrase (or cut the Abstract down even more).

5. - *The introduction is not very focused and suggest limited knowledge of clinical genetics: Why is there a focus on the difference between small /medium CNVs and large CNVs. This is not relevant for the rest of the article. Line 67: schizophrenia, ASD are placed at the same level as Williams Syndrome, a well delineated developmental syndrome. Line 72: The authors suggest that neuropsychiatric CNVs have been mostly studied by applying the paradigm of trying to determine wich single gene is important. In the field, both such CNVs are termed contiguous gene syndromes, arising from the observation that multiple genes concomitantly cause the phenotype. Rephrase. Line 78: add references. Line 97; 98: Add references (f.e. look at/within McDonald McGuinn, Nature Primer, 2015).*

We had included the part about small/medium, non-disease associated, CNVs as a means of setting the background for the large CNVs, to remind readers that these small/medium CNVs exist and to plant the idea that maybe one day it will be possible to determine molecular effects of these CNVs as well, along the lines as demonstrated by us here for the large CNV on 22q11.2. But of course we should have spelled that out more clearly while at the same time the manuscript is already very long as it is. We have therefore removed the section in question.

We have removed the reference to Williams Syndrome, that should indeed not have been there.

And we have reworded the passage about studies on single genes from within the CNV boundaries. Our original wording did indeed sound as if such studies on individual genes had been done because of paradigmatic boundaries and not of valid methodological considerations.

References were added as suggested.

6. - *It is mentioned that other studies showed an effect on gene expression outside the CNV. Refer to those studies (f.e. Harewood et al. Methods Mol.Biol.2012)*

We had indeed meant to reference the work by the Reymond group in this context. We added the reference and thank the reviewer for catching this oversight.

7. - *Results: Line 125-128: Replace this paragraph to later?*

We have modified this accordingly.

8. - Throughout the text: deletion-downstream: better to use the term deletion-distal and deletion-proximal.

We have modified this as suggested in the text.

Reviewer 3

Reviewer #3 (Remarks to the Author):

In the manuscript "Local and global chromatin interactions are altered by large genomic deletions associated with human brain development" Zhang et al. describe a Hi-C, ChIP-seq, and RNA-seq analysis of immortalized lymphoblastoid cell lines from 5 patients carrying a 22q11.2 deletion in order to study the local and global effects of such deletions on chromatin folding, epigenetic modifications and gene expression. They find that the circa 3Mb deletions mostly affect gene expression and chromatin folding at the 22q11.2 region of the genome and do not find systematic global effects on chromatin organisation. Interestingly, they show that the 3Mb deletions seem to decrease the spatial distance between the 22q11.2 region and a telomeric region of chromosome 22.

The experimental approach and generated dataset are very relevant from a medical genetics viewpoint and are promising to offer important insights into the disease etiology and molecular pathogenesis of CNVs in general and for 22q11.2 deletions in particular. However, the current amount and quality of the analysis is only a starting point for such a paper and mostly confirms previously published results at other loci (Mundlos lab for congenital disease, Korbelt lab for cancer) without providing substantial advance in our understanding of chromatin folding and disease. Aside from presenting their data in a sometimes rather difficult to follow way, what is it that we learn from this study?

We are encouraged by the assessment that this study used an approach and generated a dataset of relevance and that it promises to offer important insights into the disease etiology of molecular pathogenesis of CNVs in general and for the 22q11.2 deletion in particular.

Following the reviewer's suggestions we have expanded the experimental scope (namely by employing a novel approach allowing for haplotype-specific analysis of Hi-C data) and the computational analyses and have strived to improve the presentation of the results. We feel that there are now reporting several relevant findings, both where we show that for another important large CNV (i.e. the one on 22q11.2) there are molecular effects similar to those seen for other large CNVs, and where we show that some molecular effects are either different from those seen for other large CNVs, from what would have been expected or where we carry out analyses that have not yet been done previously for large CNVs. This will be detailed in the following, under the respective comments. Immediately below a brief summary of the findings from this study, after the revisions:

Gene expression

- downregulated within CNV boundaries (a replication of earlier studies in 22q11.2 deletion LCLs)
- differentially changed for many genes genome-wide
- affected pathways include energy metabolism (several relevant genes in 22q11.2 deletion), neurodegenerative disorders (potential relevance to 22q11DS)
- allele-specific expression analysis shows no ASE effects associated with the 22q11.2 deletion

Chromatin marks and CTCF binding

- differentially and reciprocally changed histone marks in large regions both upstream and downstream of the deletion
- differential binding of CTCF in a large region downstream (but not upstream) of the deletion
- differential changes in histone marks and CTCF binding for multiple loci genome-wide (correlating with gene expression changes)

Chromosome interactions, chromosome A/B compartments, topological domains

- intra-chromosomal interactions reduced within CNV boundaries and between CNV region and rest of 22q; and inter-chromosomally between the CNV region and the rest of the genome
- intra-chromosomal interactions increased between deletion-flanking regions
- inter-chromosomal interactions increased between downstream deletion-proximal region and a telomere-proximal region (validated by 3D FISH)
- inter-chromosomal interactions changed genome-wide, including between pairs of chromosomes other than 22q
- chromosomal A/B compartments not changed
- topological domains on intact chromosome 22q not affected; on deletion 22q the topological domain on top of the deletion is absent entirely, the flanking domains are not fusing, the upstream deletion-proximal domain increases its interactions with the first downstream deletion-proximal domain as well as with the following two domains

Integrated analyses across molecular levels

- gene expression changes and chromatin mark changes are correlated when analyzing individual genes genome-wide
- deletion-upstream and -downstream regions show decreased intra-chromosomal interactions as well as differential changes in histone mark patterns
- deletion-downstream region shows differential change in CTCF binding as well as increased interactions with telomere-proximal region

- deletion-upstream and -downstream regions as well as telomeric interaction region contain differentially expressed genes but are not enriched for them

Furthermore there is replication of a limited set of the above findings in the 1q21.1 CNV.

We think our study was important and useful also because of the prominence and particular nature of the 22q11.2 deletion, which has a higher occurrence compared with other large structural variants and where the patients show very striking combinations of specific neuropsychiatric disorders and other developmental phenotypes. For instance, individuals with the 22q11.2 deletion have a 20- to 30-fold increased risk for schizophrenia, making it the largest genetic risk factor for schizophrenia. Yet the molecular foundations of the etiology of 22q11DS are still only very poorly understood, in particular regarding the neurodevelopmental and neuropsychiatric phenotypes.

To improve our manuscript and address the reviewer’s criticism, we performed additional analyses, and also additional experiments by carrying out capture Hi-C and haplotype phasing for two patient and two control cell lines. With these data we were able to gain for the first time haplotype-specific insights into the effects of the 22q11.2 deletion on several levels of molecular activity (please see more details below).

1. The authors touch at least two interesting points that could reveal interesting biology if followed up in more detail; the switch of A/B compartments and the interaction between very distal parts of chr22. For the A/B switch: How reproducible is this? Is this seen in every patient cell line and every control cell line? Does this correlate with called regions of differentially regulated genes and chromatin marks?

We thank the reviewer for these suggestions, we can indeed clarify this much more. A/B compartments as shown in the original Fig. 3a were calculated by averaging the interactions across all the individuals within the control and patient groups separately. The compartment switch was reproducible in each cell line, although sequencing depth seems to have an impact (GM18505 and GM06990 have less sequencing reads than other individuals). The patient cell lines are as follows:

The control cell lines are as follows:

The difference of PC1 values between controls and patients only occurred in the deletion region, where there was a decrease of gene expression and chromatin marks signals (Fig. 1e-h) in patients. But we did not see the change of PC1 values in other regions enriched with differentially expressed genes and differential chromatin marks.

As reviewer 2 mentioned above in comment #1 and reviewer 3 asked below in comment #3, we agree entirely that this change of PC1 values was only the reflection of the two signals from two chromosomes combined and could not distinguish the signal from the intact chromosome from that with deletion without carrying out haplotype phasing first. Thus, we removed the original Fig. 3a and Fig.

7c from the text. However, inspired by the questions of the reviewers we decided to phase two controls and two patients and perform deep capture Hi-C to pursue the answer in an haplotype-specific way (please see details below and also in the Methods and Supplementary Information). It turned out the intact chromosome and the deleted chromosome had the same sign of PC1 values outside the deletion region, indicating no A/B compartment change occurred on the deleted chromosome 22q. The change of PC1 in the deletion region using the unphased data was only due to the decrease of interactions in patients and thus not a real reflection of the A/B compartments.

2. The very long range interaction. Is there a large structure of chromosome 22 in which centromeres and telomeres of the chromosome are in close chromosomal contact? The deletion would bring RP11-47L18 very close to the centromere. Or what is the molecular relevance of this chromosome-scale observation? Does this affect gene expression?

We looked into our newly generated capture Hi-C data to check if the centromeres and telomeres of chromosome 22q are in close proximity to each other. Below are examples of normalized contacts of chromosome 22q (red is more interactions than expected, blue is less; the yellow box indicates the region of contact between centromere and telomere):

In the heatmaps above we observe that the contacts between the centromere and telomere of chromosome 22q (yellow box) are not more intense than expected. Therefore our data does not support that centromeres and telomeres of chromosome 22q are generally in close proximity to each other. This was consistent with the similar heatmap (please see the figure below) from the original Hi-C paper (Lieberman-Aiden et al. Science. 2009).

Excerpted from Figure S1 of Lieberman-Aiden et al. Science. 2009.

There are two genes that are differentially expressed between patients and controls, out of 24 genes within the telomeric 50-51 Mbp region. But the number of genes with differential expression was not enriched (Fisher's exact test $p = 0.5653$) within this region compared to the whole genome background (1,610 genes differentially expressed out of 11,374 genes). We discussed this in the third paragraph of the Discussion (lines 538-540 in the revised manuscript or lines 641-643 in the changes tracked version).

Aside from this, there are additional issues where the data analysis seems incomplete:

3. - *The authors importantly state the problem of normalization in heterozygous deletion alleles and overcome this problem by using the hicpipe algorithm. For a fully informative analysis of the data, the interaction on the allele carrying the deletion should be at least be approached, e.g. by subtracting half the wildtype signal from the mutant, thereby providing a view of the "mutant allele only". I agree that haplotype-phasing would be the optimal, but very difficult thing to do and would warrant a publication in its own right.*

We entirely agree with the reviewer on this. The optimal and only accurate way to determine the interactions of the deleted chromosome 22q separate from those of the intact 22q is to carry out haplotype phasing and then map only such Hi-C reads that carry informative SNVs on top of the phased genome maps. We had indeed planned this as a follow-up study, to become a separate publication, as mentioned by the reviewer.

However, inspired by the questions raised by the reviewers, we decided to tackle the difficult problem of haplotype specific Hi-C analysis already now in order to complement and significantly improve the current manuscript. We conducted haplotype phasing on two patient cell lines and two control cell lines. Also, to generate Hi-C sequencing coverage deep enough to contain the necessary number of reads containing informative, heterozygous, SNVs, we designed oligomer capture probes targeting the entire sequence of chromosome 22q and performed capture

Hi-C on these genome-phased cell lines. We think in this combination this is a novel approach and it required some development on both the experimental and computational sides.

To phase the haplotypes we used a method called statistically aided, long-read haplotyping (SLRH), previously reported (Kuleshov et al. Nat. Biotechnology. 2014), to generate haplotype contigs with lengths of 0.2–1 Mbp (detailed in the Methods and Supplementary Information). Briefly, genomic DNA was sheared into fragments of about 10 kbp, diluted and placed into a 384-well plate. Each well contains 3,000–6,000 molecules to keep unique DNA sequences at a low frequency so that the fragments within each well will almost never come from the same locus, allowing haplotypes to be assembled after sequencing. Within each well fragments were amplified, cut into short fragments and barcoded with a unique barcode for each well. Fragments from all wells were pooled together and sequenced. Sequencing reads were mapped to reference genome and assigned to their original well based on the barcode specific to each well. Within each well, reads were assembled at their overlapping heterozygous SNVs into haplotype blocks, which were assigned to long haplotype contigs statistically based on a phased reference panel using the haplotyping algorithm Prism developed by Illumina.

However, these haplotype contigs need to be connected in order to distinguish the two homologous chromosomes in full length. The strategy we employed to address this issue was to use two patient cell lines from related donors (ID00014 is a parent of ID00016) and two control cell lines from related donors (GM12892 is a parent of GM12878). The haplotype contigs can be assigned to their proper chromosome using the phased heterozygous SNVs whose alleles' origin can be inferred based on Mendelian inheritance.

Take ID00016 as example. We were able to tell which one of the two alleles of a SNV that is heterozygous SNV in ID00016 but homozygous in ID00014 was inherited from ID00014 and which one was not. If a haplotype contig of ID00016 contains such a SNV it would enable us to tell if this haplotype contig was inherited from ID00014 or not. We then connected all the haplotype contigs of ID00016 inherited from ID00014 to obtain the whole chromosome inherited from ID00014 and all the ones not inherited from ID00014 to obtain the whole chromosome not inherited from ID00014.

The interactions of the chromosome 22q with deletion for ID00014 and ID00016 are shown in the new Fig. 2, together with the interactions of the intact chromosome 22q, and also the haplotype-specific interactions for each chromosome 22q separately for the two control cell lines.

4. - Hi-C data: Similarities and differences between deletion cell lines (Figure 1 and 3). Have the authors always combined the Hi-C data from 5 patient cell lines to perform their analysis? How do the alleles differ and why do they need to/why should they be combined in the analysis? How reproducible between individuals are detected contact and gene expression changes.

We only combined the Hi-C data for patients and controls, respectively, for calling topological domains in order to achieve enough resolution. All the other analyses in the manuscript were done on each cell line separately and the comparisons between patients and controls were performed by then comparing the two groups of cell lines.

The contact changes observed in patients are reproducible for each patient. The heatmaps of chromosome 22q's interactions are shown as follows (on the left side are patients while on the right side are controls; the deletion region is marked by black lines):

The gene expression changes in the patients are also reproducible for each individual. An IGV snapshot showing the FPKM values for each gene on chromosome 22q for each individual is shown below (red colors are patients while blue colors are controls; the deletion region is marked by black lines):

5. As shown in Figure 1a the deletion creates contacts between the now adjacent parts of the genome. The findings from the Mundlos lab would predict that these contacts extend only to the end of the next TAD or some other similar structure. Is this the case? Do we get a new, fused TAD between the upstream and downstream regions of the deletion? The resolution in Fig. 1a is 500kb and cannot be used for this and the interaction map in Figure 3b actually cuts out this highly informative piece of information and from what is presented, the data looks very sparse. Also genes, gene expression and ChIP-seq tracks should be included in such a figure.

We have found these comments very helpful. From the comparison of the heatmaps of the chromosome 22q with deletion and intact chromosome 22q in the patient cell lines we find that the increased contacts extended to multiple TADs downstream of the deletion rather than only to the end of the next TAD or the one after next due to the variations of the TAD calling (Fig. 4, Supplementary Fig. 6-7; also please note that we refer to TADs as topological domains in the text overall and in the following). The boundaries of the original topological domains were not influenced by the deletion. The increased contacts end at the boundary of the last topological domain involved.

We did not see a newly fused topological domain between the upstream and downstream regions of the deletion despite the increased contacts of the topological domains upstream and downstream of the deletion. One could speculate that these increased contacts were either not strong enough or not of the right nature to prompt the deletion-proximal topological domains to be fused into a new topological domain.

We removed the original Fig. 3 and presented the haplotype-specific contacts in the newly added Fig. 2 and the haplotype-specific topological domains in the newly added Fig.4 as well as in Supplementary Fig. 6 and Supplementary Fig. 7, using the

newly generated capture Hi-C data of chromosome 22q (please see more details in the answer to the next comment and also the Methods) at 200 kbp resolution. We added tracks for gene expression and ChIP-Seq data in these newly added figures as suggested by the reviewer, in a haplotype-specific manner.

6. - *If the sequencing depth is not sufficient to create maps of higher resolution (which might possible, see Sup Tab. 5) it might make sense to increase sequencing depth for just one or two individual experiments to create a high-res maps, that allow a better view of the TADs than what is currently presented.*

We followed this suggestion and increased the sequencing depth for the Hi-C libraries of two patient cell lines and one control cell line by adding one and a half lanes of HiSeq4000 run, using 2x150 bp paired-end sequencing (read output numbers were 937, 1,017 and 848 million for ID00014, ID00016 and GM12878 respectively). Combining the original data and the new data brought the total reads number to 1.7 billion reads for each of these three cell lines. The combined data markedly improved the resolution of the unphased contacts maps.

However, when we sought to investigate the topological domains in a haplotype-specific way, as suggested by the reviewers, we found that the Hi-C sequencing coverage was still not high enough, in spite of the already large number of reads now available. The resolution of haplotype-specific contact maps was not very high as only reads mapped to phased heterozygous SNVs could be used when constructing contact heatmaps for each homologous chromosome. Below is a plot of haplotype-specific contacts of the two homologous chromosomes 22q for GM12878 at 200 kbp resolution using this data:

To obtain Hi-C sequencing coverage for chromosome 22q of a depth that would allow for haplotype-specific analysis at satisfactory resolution we decided to perform chromosome-wide capture Hi-C on newly generated in situ Hi-C libraries for the four cell lines with haplotype phasing data. Oligomer capture probes forming

a dense tiling path along chromosome 22q were designed using NimbleDesign. Roche-NimbleGen's SeqCap EZ Choice XL Enrichment Kit was used to capture Hi-C fragments with at least one end from chromosome 22q from the in situ Hi-C libraries. With this capture Hi-C data we achieved enrichments of 10-16 fold for chromosomal contacts involving chromosome 22q, compared with genome-wide Hi-C data. Below is a genome-wide plot of Hi-C contacts for GM12878 using the capture Hi-C data (plots for all the four cell lines were added as Supplementary Fig. 3).

From the plot, we can see clear enrichment of intra- and inter-chromosomal contacts of chromosome 22, indicating the capture Hi-C worked as expected. There are also Hi-C data for all the other chromosomes, which is a result of the NimbleGen capture process being somewhat “leaky”. But the vast majority of reads are concentrated on chromosome 22q, as desired. In this fashion we generated 700-800 million chromosome 22q specific capture Hi-C sequencing reads for each cell line. To achieve the same coverage on 22q with genome-wide Hi-C would have required 7-12 billion Hi-C sequencing reads for each cell line (i.e. several entire Illumina flow-cells for each cell line). The newly generated haplotype-specific heatmap of contacts are shown in the new Fig. 2. Below is the plot of GM12878 for comparison with the unphased data above.

7. - Chromatin marks and gene expression: The authors find that H3K27ac and gene expression in the 22q11.2 flanking regions are affected. How do the decreased H3K27ac and increased H3K27me3 marks correspond to the downregulated *BID* and *IPLL5* genes? Are these individual enhancers, only the promoter of the gene etc.? Only showing the 500kb binned data is not sufficient.

We thank the reviewer for these questions. We redid the analysis on a gene-by-gene basis. We checked the peaks of H3K27ac and H3K27me3 for *BID* and *IPLL5*. For *BID*, there was no difference in the H3K27me3 marks in either the promoter or the enhancers. We found increased H3K27ac marks in the promoter and the enhancers in the patient cell lines, which is the opposite of the expected given the downregulated expression of *BID* in the patient cell lines. Below is the plot of gene expression and the chromatin marks (green are patient cell lines while blue are controls; regions in red are differential binding sites of H3K27ac; ChIP-Seq and RNA-Seq read counts were normalized to 10 million read pairs and scaled to the same level for each dataset):

For *IGLL5*, we saw decreased H3K27ac marks and increased H3K27me3 marks in one of the enhancers in the patient cell lines. There were also increased H3K27me3 marks in the promoter region. These changes were consistent with the downregulated gene expression of *IGLL5* in the patient cell lines. Below is the plot of gene expression and of the chromatin marks (green are patient cell lines and blue are controls; regions in red are sites with differential H3K27ac marks; ChIP-Seq and RNA-Seq read counts were normalized to 10 million read pairs and scaled to the same level for each dataset):

Taken together, the downstream deletion flanking region exhibited a consistent change of chromatin marks and gene expression while the upstream deletion flanking region did not. In our view this rendered the association between gene expression and chromatin marks in the deletion flanking regions inconclusive.

Therefore we removed the two paragraphs in the Results section (lines 311-324 in the changes tracked version) and the lines in the Discussion section (lines 630-638 in the changes tracked version) that were referring to the consistent changes between gene expression and chromatin marks that were based on the analysis using 500 kbp binned data to avoid.

8. Moreover, it should be possible to use the SNP information in Hi-C and RNA-seq data to assess the allele-specific changes in gene expression (i.e. expression from the “wildtype” allele should be unaffected, the 22q11 del-allele should have weaker expression?)

We again thank the reviewer for another constructive and helpful comment. With the haplotype phasing data in hand we were able to perform allele-specific expression (ASE) analysis (details in the Methods) on the whole chromosome level, first on the four haplotype-phased cell lines. We did not observe any gene on chromosome 22q which only exhibited ASE in patient cell lines but not in control cell lines. Below is a plot for the expression of two homologous 22q chromosomes for the four cell lines with haplotype phasing data (also shown as Supplementary Fig. 13; green are patient cell lines, blue are controls; RNA-Seq read counts were normalized to 10 million read pairs and scaled to the same level; read counts of the 22q11.2 region of the intact chromosome 22q in ID00016 and ID00014 were shown at the same positions where there are informative SNVs in GM12878 or GM12892, respectively):

We then expanded our ASE analysis to all the other cell lines without haplotype phasing data. Consistently, we did not identify any gene on chromosome 22q only exhibiting ASE in patient cell lines but not in control cell lines. Taken together, our data did not provide evidence of differential expression between the intact chromosome 22q and the chromosome 22q with deletion in the LCLs in this study.

We now mention this in the Discussion section (lines 542-545 in the revised manuscript or lines 645-648 in the changes tracked version).

9. - Analysis of ChIP-seq data (Figure 2): With ChIP-seq the authors have potentially enhancer-level information on histone modifications (see also above). However, they decide to analyse it by plotting the p-value of differentially enriched marks in 500kb windows, not giving information how they relate to wildtype or newly formed TADs, compartments or differentially expressed genes. What are we supposed to learn from these graphs?

We understand the criticism and wish to clarify the analyses we did for this part of the study. The analysis of differential histone modifications and CTCF binding was based on individual ChIP-Seq peaks. To investigate how histone modifications relate to differentially expressed genes, we assigned the histone modification peaks to their associated genes based on the distance to their transcription start sites and conducted various association analyses. We observed that gene expression changes correlated well with histone modification changes. The results were shown in the section “Correlation between histone modification and gene expression” in the Results (lines 436-473 in the revised manuscript or lines 519-557 in the changes tracked version). These analyses were based on individual genes and individual histone modification peaks rather than in 500 kbp windows.

The suggestion by reviewer 3 in comment #7, adding tracks for histone modifications and gene expression in the original Fig. 3b, was very helpful. However, as both reviewer 2 and 3 had pointed out, the originally reported signals for topological domains and A/B compartments were computed from a mixture of data from two homologous chromosomes 22q. After generating haplotype specific analyses for the topological domains and A/B compartments, it was implausible to show those haplotype specific results in combination with non-haplotype specific ChIP-Seq signal tracks.

Based on the newly generated capture Hi-C data and haplotype phasing data we were able to build haplotype-specific A/B compartments and topological domains for both the intact and deletion-carrying chromosome 22q. Our data showed that the A/B compartments remained unchanged on the chromosome 22q with deletion compared with the intact one. The boundaries of the topological domains also remained unchanged. There were no newly formed topological domains on the chromosome 22q with deletion while the interactions between the one topological domain upstream and the three topological domains downstream of the deletion were found to be increased.

We next performed haplotype-specific ChIP-Seq analysis for the 3 cell lines using the original both ChIP-Seq data and the newly generated haplotype phasing data and looked into whether there was a difference between the intact chromosome 22q and the one with deletion. In the upstream region (18-18.5 Mbp) of the 22q11.2 deletion, we did observe that H3K27ac marks were decreased and H3K27me3 marks were increased on the chromosome 22q with deletion in the

regions where there are both differential H3K27ac and H3K27me3 marks between patients and controls (please see the figure below; tracks in black indicate the boundaries of the peaks and the fold change of patients vs. controls).

However, in the deletion-downstream region (23-23.5 Mbp) we did not see consistent changes of histone modifications between the intact chromosome 22q and the one with deletion that were shared by both patients (please see the figure below).

Taken together, it seemed that the increased contacts of the two deletion flanking regions of the 22q11.2 deletion on chromosome 22q with deletion only affected the histone modifications in the upstream flanking region but not the downstream flanking region. The histone modification changes in the downstream flanking region occurred on both the intact and deletion-carrying chromosomes in the patients, which might not be directly caused (at least not only) by the increased

contacts of the flanking regions of the deletion but could be driven by gene expression network changes and perhaps also contacts involving other regions. In the meantime, we think that while these particular results are very interesting, they are not solid enough for inclusion in the manuscript as they are based on a small sample size (2 patients and 1 control) and the heterozygous SNVs in the called peaks vary among individuals. Therefore, we only view these particular findings as very preliminary results and further work is required to confirm or refute them in larger samples and in more relevant tissues associated with the disease phenotypes.

We hope that this revised manuscript that includes a considerable amount of additional data and analyses would address the comments and suggestions made by the reviewers. We feel that by working our way through these extensive comments and suggestions has improved the manuscript very markedly and we again thank the reviewers for this.

Reviewer #1 (Remarks to the Author):

The authors have addressed my questions adequately.

Reviewer #2 (Remarks to the Author):

The authors did address the main concerns raised during the first review. More specifically, the haplotype based 4C analysis is an impressive improvement to the manuscript. This analysis has changed many of the original claims made and makes the findings more relevant.

Considering this is a technological innovation

Minor comments:

- Line 270: The figure legend state 'the gap... is caused by a very low density of heterozygous SNPs'. But since the plots are haplotype specific, there are no heterozygous SNPs. I would think that this is the deletion?
- Line 675: Maybe refer to Gur et al. (28761081)

Reviewer #3 (Remarks to the Author):

I have now read the revised manuscript NCOMMS-17-10901A by Zhang et al.. I appreciate the extensive effort the authors made to address the criticism raised by the reviewers. Again, I would like to state that I think the idea and experimental setup is good and the study would contribute to elucidating the molecular effects of CNVs. I especially appreciate the efforts made to generate haplotype-phased capture Hi-C maps.

In my original review I highlighted two interesting observations that – if followed up in more detail – would make an interesting story; “the switch of A/B compartments and the interaction between

very distal parts of chr22". After re-evaluation of their Hi-C data the authors found that the identification of the A/B compartment switch was identified due to the problematic analyses. Regarding the long-range interaction, the authors do not provide any further details aside from what was reported in the original manuscript.

This leaves the manuscript with the result that in 22q11 patient LCLs global chromatin contacts and gene expression are systematically reduced only for the deleted region, supported by the fact that histone modifications (of the TSSs) correlate with changed gene expression, as one would expect from numerous previous studies.

The haplotype-phased capture Hi-C of chr22 is a new and interesting point of the manuscript, however, this brings me to my biggest concern regarding the presented data; the quality of the Hi-C experiments. In my original review I commented on the Hi-C data and also alluded to the sequencing depth and the problems that these pose with calling topological domains and presenting the data.

To be more explicit, in Supplemental Tab. 6 the authors list the number of cis and trans contacts pre- and post-filtering (i.e. the removal of PCR duplicates (ll. 662)). The post-filtering cis/trans ratio is between 0.19 and 0.39, which leaves the authors with very sparse data and as little as 5.5 mio contacts (average 13mio) to produce a Hi-C map for an individual (e.g. ID00016 map is missing data over several Mb in the 22q13.1-2 region). Even with the pooled data from all patients/controls, only about 70 million reads are available to produce a cis-contact map for all human chromosomes. This is very little to produce genome wide Hi-C maps or to make any sort of statements about topological domains (as is done in ll. 288) but more importantly hints at the fact that something at the wet-lab side of the experiments might have been sub-optimal.

I cannot find according data pre/post filtering numbers for the capture Hi-C data that was generated with the in-situ Hi-C protocol and might not have the same systematic biases. However, the data presentation in Figure 2 indicates that the data quality might be impaired as well.

As an example, the cis-to-trans ratio in the Supplemental Table 1 from the 2012 Hi-C paper by the Ren lab (Dixon et al. 2012) shows that in order to produce good quality Hi-C maps and analyze cis-contacts (such as those on chromosome 22), a higher proportion of cis-contacts versus trans-contacts should be expected from a Hi-C experiment. Because of the better cis/trans ratio, the number of cis-reads in the least covered replicate of Dixon et al. (hESC, Original ~50Mio cis contacts) is about the same as for all pooled experiments in this manuscript.

Also the more detailed and informative breakdown into short and long-range cis-contacts is missing from Supplemental Tab. 6, which is informative, as the Hi-C signal is mainly generated by the >20kb intra-chromosomal contacts.

Supplemental Table 1 from Dixon et al. 2012 (see attachment)

The sub-optimal quality of the primary Hi-C data does not devalue all analyses of the manuscript at hand, but produces a very low resolution Hi-C experiment and very much limits the possible analyses to physical separation at A/B compartment level and general interaction frequencies across the genome.

In the revision process the authors nicely show that the reported switch in A/B compartments from the original version of the manuscript was due to a problem with the data analysis.

Also, I am wary to believe the global changes in trans-contacts (Fig. 6 and S12), although tested with permutation tests, until replicated with higher-quality Hi-C data.

The authors state that the TADs adjacent to the microdeletion do not change. As mentioned above, I find this very hard to extract from the Hi-C data generated in this study, however, I do believe that the authors are correct with their statement. The 22q11 deletion region is located in between several clusters of repeats/microsatellites that are a) difficult to map onto a reference human genome and b) likely the breakpoints for the recurrent deletions at this locus. Consequently, even in control individuals these microsatellites are possibly some sort of boundaries of flanking new topological and/or regulatory domains and the flanking domains should be not affected. High-quality Hi-C experiments in combination with the already generated ChIP-seq data would help to study this in detail.

I would refrain from an epigenetic analysis for differentially enriched ChIP-seq signals as performed in Figure 3a-c. As the authors point out correctly in their reply to my earlier comments, individual regions/enhancers/promoters can behave differently from one another and make the analysis “inconclusive”. Plotting the \log_{10} -pvalue of \log_2 -transformed fold-changes in “[500kb] bins with significantly differentially enriched sites against the background of the whole genome.” (lines 888-895) will produce a result that will not reflect what happens in the deletion flanking regions on an epigenetic level.

All the reader can take from such an analysis is that there is some sort of change in the regions flanking the breakpoints, but essentially the reader is left alone with the interpretation with no possibility to evaluate the data themselves.

In the below please find our point-by-point response to the Reviewers' comments.

Reviewer #1 (Remarks to the Author):

The authors have addressed my questions adequately.

We thank the Reviewer again for the very helpful comments and remarks earlier during the review process, which contributed substantially to improving this manuscript.

Reviewer #2 (Remarks to the Author):

The authors did address the main concerns raised during the first review. More specifically, the haplotype based 4C analysis is an impressive improvement to the manuscript. This analysis has changed many of the original claims made and makes the findings more relevant.

Considering this is a technological innovation

We appreciate the Reviewer's comments. We also think the haplotype-specific capture Hi-C analysis did significantly improve the original manuscript and we are very glad that the Reviewers strongly encouraged us to undertake this additional analysis.

We also consider the haplotype phasing of the Hi-C signals as a general technological innovation, besides of the findings its use yielded in 22q11 Deletion Syndrome. We combined the cutting-edge approach of statistically-aided long-read haplotyping (SLRH) with an analysis of Mendelian inheritance of single nucleotide variants (SNVs) and then used the resulting haplotypes for phased analysis of very-deep chromosome-wide capture-Hi-C data. We hope that this approach will prove useful to other researchers who are analyzing the effects of CNV or other structural genome variation on chromosome folding patterns.

Minor comments:

- Line 270: The figure legend state 'the gap... is caused by a very low density of heterozygous SNPs'. But since the plots are haplotype specific, there are no heterozygous SNPs. I would think that this is the deletion?

We would like to clarify that the 22q11.2 deletion region is from 18.8 Mb to 21.8 Mb, (e.g. indicated by the dashed blue lines in Figure 2). By "The gap in signal for the region from 39 Mbp to 42 Mbp in ID00016 is caused by a very low density of heterozygous SNVs", we meant the 22q13.1-2 region of ID00016's heatmap in Figure 2a. Only regions with sufficiently high numbers of heterozygous SNVs can be phased with the combined linked-read/familial genotypes based approach we were using. This requirement is leaving a gap in the phasing in the region from 39 Mbp to 42 Mbp in ID00016. However, while this region is on chromosome 22q it is far distant from the 22q11.2 deletion region and was not a region of interest in our study, also when looking across our entire cohort. The lack of phasing in this region does not affect our integrative analyses or findings in general.

- Line 675: Maybe refer to Gur et al. (28761081)

Thank you for the suggestion, we added this reference.

Reviewer #3 (Remarks to the Author):

1. I have now read the revised manuscript NCOMMS-17-10901A by Zhang et al.. I appreciate the extensive effort the authors made to address the criticism raised by the reviewers. Again, I would like to state that I think the idea and experimental setup is good and the study would contribute to elucidating the molecular effects of CNVs. I especially appreciate the efforts made to generate haplotype-phased capture Hi-C maps.

We are very thankful for these positive comments. The Reviewer had pointed out in the first round of the review that the haplotype-phased capture Hi-C analysis would be very interesting to look at, but also that it would be a challenging thing to undertake that might even warrant a separate publication. However we were motivated by this comment (as well as the similar comment by Reviewer 2) to attempt the phased analysis already as part of the current manuscript. We are glad that we did and again we thank the Reviewers for motivating us to do so. We believe the resulting current manuscript is presenting much more advanced findings regarding the effects of the 22q11.2 deletion on epigenomic levels of regulation. The 22q11.2 deletion is one of the strongest genetic risk factors for several psychiatric disorders and probably one of the most prominent disease associated large CNVs in humans.

Also we would like to thank the Reviewer for prompting us, in this 2nd round of revisions, to carry out comparative analysis of QC metrics, between our Hi-C data and that of several landmark Hi-C publications (Reviewer comment 4., below, and our analysis following that comment). This comparative analysis of QC metrics demonstrates that our Hi-C data is of a quality on par with the data from several Hi-C landmark papers, as a result of which we can now have an additional level of confidence in our findings based on that data.

2. In my original review I highlighted two interesting observations that – if followed up in more detail – would make an interesting story; “the switch of A/B compartments and the interaction between very distal parts of chr22”. After re-evaluation of their Hi-C data the authors found that the identification of the A/B compartment switch was identified due to the problematic analyses. Regarding the long-range interaction, the authors do not provide any further details aside from what was reported in the original manuscript.

We would like to clarify that the analyses that lead to the signal indicating the switch in A/B compartments around the 22q11.2 deletion locus were not problematic in the sense that they were done the wrong way – but rather these analyses were done according to the state-of-the-art, using standard procedures in the Hi-C field, namely procedures that do not distinguish between haplotypes but rather simply conflate the signals for a given locus coming from the two homologous chromosomes. This shows that the current state-of-the-art in the Hi-C field when applied to large CNVs is incomplete and our manuscript now, thanks to the prompting of the Reviewers, describes a remedy to this problem, which we would posit makes it of particular interest to the field, beyond the questions specifically surrounding the 22q11.2 CNV. We feel that therefore the absence of the A/B switch in the revised manuscript is a function of a methodology advance that should be of general usefulness to the chromosome conformation field as a whole.

Regarding the change in long-range interaction that we reported in the original version of the manuscript, between the downstream deletion proximal region and a telomere-proximal

region, we did not provide further analyses or details in the revision as we posit that the original observation is of significant interest as described, and the finding is already strongly supported by the experimental validation done using an orthogonal method, 3D FISH. We agree that this is one of the exciting observations in our manuscript and the next questions following from it will be, for example, what are the molecular mechanisms that mediate this change in long-range interaction, and how are these mechanisms affected by the large CNV. These questions will be the subject of an entire and extended follow-up study, or studies, after this observation has been reported to the field. 3D FISH is already a very established, broadly-accepted, if also laborious and low-throughput, method to validate chromosome interactions, and probably the gold standard for this purpose. Therefore, for this particular observation, again, we submit that no further validations are necessary.

There is an additional point to be made regarding the 3D FISH validation of long-range interaction changes predicted by Hi-C. Namely, the ability to validate predictions coming from the Hi-C analysis, with an orthogonal and sequencing-independent method such as 3D FISH, is a further strong sign of the high quality of the Hi-C data that is underlying the analysis (i.e. regarding Reviewer comment number 4., below).

3. This leaves the manuscript with the result that in 22q11 patient LCLs global chromatin contacts and gene expression are systematically reduced only for the deleted region, supported by the fact that histone modifications (of the TSSs) correlate with changed gene expression, as one would expect from numerous previous studies.

The chromosome contacts, gene expression levels and epigenetic signatures within the 22q11.2 deletion boundaries were indeed reduced in the patient lines as described above by the Reviewer. However, our manuscript also reports an entire catalog of additional findings, extending beyond the findings limited to within the deletion boundaries. Below is a brief summary, of this catalog of novel findings outside the 22q11.2 deletion boundaries. These are just the findings from the initial manuscript, to which we then added the haplotype-specific findings (and the approach of generating these haplotype-specific findings as a novel method) in the revised manuscript.

Gene expression

- differentially changed for many genes genome-wide
- affected pathways include energy metabolism (several relevant genes within the 22q11.2 deletion), neurodegenerative disorders (potential relevance to 22q11DS had previously been reported on the clinical level)
- allele-specific expression analysis shows no ASE effects associated with the 22q11.2 deletion (this information may prove useful during the follow-up studies into molecular mechanisms controlling these gene expression changes – whatever these mechanisms are, they affect both alleles equally)

Chromatin marks and CTCF binding

- differential and reciprocal changes in activating and repressing histone marks in large regions both upstream and downstream of the deletion
- differential binding of CTCF in a large region downstream (but not upstream) of the deletion

- differential changes in histone marks and CTCF binding for multiple loci genome-wide (correlating with gene expression changes for these loci genome-wide)

Chromosome interactions

- intra-chromosomal interactions increased between deletion-flanking regions
- intra-chromosomal interactions increased between downstream deletion-proximal region and a telomere-proximal region (validated by 3D FISH)
- inter-chromosomal interactions changed genome-wide, including between pairs of chromosomes other than 22q

Integrated analyses across molecular levels

- gene expression changes and chromatin mark changes are correlated when analyzing individual genes genome-wide
- deletion-upstream and -downstream regions show increased intra-chromosomal interactions as well as differential changes in histone mark patterns
- deletion-downstream region shows differential change in CTCF binding as well as increased interactions with telomere-proximal region
- deletion-upstream and -downstream regions as well as the telomeric interaction region contain differentially expressed genes (but are not enriched for them)

The literature is only very sparse regarding integrated genomics/epigenomics studies of the molecular effects of the large CNVs that are associated with psychiatric disorders (e.g. the CNVs on 22q11.2, 16p11.2, 15q13.3, 1q21.1, 15q11.2, 2p16.3, or 3q29), in stark contrast to the importance of, and interest in, these large CNVs. Rees *et al.* 2014 (PMID: 24217254) reported on the gene expression effects of 22q11.2 deletions and duplications in LCLs and Blumenthal *et al.* 2014 (PMID: 24906019) studied the gene expression effects of 16p11.2 deletions and duplications, also in LCLs. Neither of these studies investigated any epigenetic changes, let alone haplotype-specific chromosome conformation changes. Loviglio *et al.* 2017 (PMID: 2724053) performed 4C-seq in LCLs with 16p11.2 deletions or duplications to study chromosome conformation changes directly involving that large CNV, without haplotype-specific analysis and with only limited integration with other epigenetic data (data, such as ChIP-Seq, that had not been generated from the LCLs actually carrying the large CNV). No studies have been reported about the chromosome conformations and epigenetic profiles of 22q11.2 deletion patients. No previous publications have studied the CNVs in a haplotype-specific way, an approach that can be expected to be impactful beyond its application to 22q11.2. The size of a given CNV and its position relative to topological domain-defining sequence elements might determine its effects. Therefore we consider differences between findings in our study of cells with the 22q11.2 CNV, and, for example, the important work of the Mundlos group on other CNVs, particularly interesting and worthwhile reporting. Such differences do not indicate that there are problems with one or the other study but just show that we need to study the effects of all major CNVs, one by one, on chromosome conformation, to eventually be able to distill general rules from the combined results of such studies. Looking at this perspective from a different angle one could say that even if every finding in our study for 22q11.2 had already been observed in other large CNVs (which it hadn't), it would still have been important to also investigate these aspects for the 22q11.2 CNV, one of the major large CNVs with disease association in humans.

4. The haplotype-phased capture Hi-C of chr22 is a new and interesting point of the manuscript, however, this brings me to my biggest concern regarding the presented data; the quality of the Hi-C experiments. In my original review I commented on the Hi-C data and also alluded to the sequencing depth and the problems that these pose with calling topological domains and presenting the data.

To be more explicit, in Supplemental Tab. 6 the authors list the number of cis and trans contacts pre- and post-filtering (i.e. the removal of PCR duplicates (ll. 662)). The post-filtering cis/trans ratio is between 0.19 and 0.39, which leaves the authors with very sparse data and as little as 5.5 mio contacts (average 13mio) to produce a Hi-C map for an individual (e.g. ID00016 map is missing data over several Mb in the 22q13.1-2 region). Even with the pooled data from all patients/controls, only about 70 million reads are available to produce a cis-contact map for all human chromosomes. This is very little to produce genome wide Hi-C maps or to make any sort of statements about topological domains (as is done in ll. 288) but more importantly hints at the fact that something at the wet-lab side of the experiments might have been sub-optimal.

I cannot find according data pre/post filtering numbers for the capture Hi-C data that was generated with the in-situ Hi-C protocol and might not have the same systematic biases. However, the data presentation in Figure 2 indicates that the data quality might be impaired as well.

As an example, the cis-to-trans ratio in the Supplemental Table 1 from the 2012 Hi-C paper by the Ren lab (Dixon et al. 2012) shows that in order to produce good quality Hi-C maps and analyze cis-contacts (such as those on chromosome 22), a higher proportion of cis-contacts versus trans-contacts should be expected from a Hi-C experiment. Because of the better cis/trans ratio, the number of cis-reads in the least covered replicate of Dixon et al. (hESC, Original ~50Mio cis contacts) is about the same as for all pooled experiments in this manuscript.

We would like to thank the Reviewer again for prompting us, in this 2nd round of revisions, to carry out comparative analysis of QC metrics, between our Hi-C data and that of several landmark Hi-C publications. As we show in the following with detailed comparative QC analyses, there is no reason for concern about the sufficient quality of our Hi-C data or quality control during the experimental portion of the study. And, again, as a result of this comparative QC analysis we can now have an additional level of confidence in our findings based on that data, therefore this comparative QC analysis was a very useful suggestion.

We describe now all the details of how our data was processed, including standard steps for Hi-C data processing, which we had not reported in detail previously. We think this will clear up any potential uncertainty regarding the comparison of QC metrics between our data and data from landmark Hi-C studies from the literature.

We downloaded and carefully parsed the data from the Dixon et al. 2012 study (available from GEO, under accession number GSE35156) that the Reviewer mentions. We would like to clarify that, importantly, the number of *cis*- and *trans*-contacts in Supplemental Table 1 from Dixon et al. 2012 were calculated from partially filtered data after removing only non-mapping reads and PCR duplicates. Therefore the Dixon et al. numbers should be used only in comparison with our “Raw_cis #” and “Raw_trans #” values in the original Supplementary Table 6. The numbers in these two columns were calculated from the partially filtered data in our dataset, after only removing the non-uniquely mapped reads and PCR duplicates. Because removing non-

uniquely mapped reads and PCR duplicates are very basic steps and routine in the analysis of the DNA sequencing data, we had simply labeled the columns which showed the number of contacts after these steps as “Raw_cis #” and “Raw_trans #”, without further elaborating how the data reflected in these columns had been treated. The question about data quality thus arose to some degree probably because of our omission of a detailed description of the basic steps used for processing the data that was shown in our Supplementary Table 6. We have now added a detailed legend for Supplementary Table 6 and also a sentence in line 665-667 of the text in the revised version.

There were four human samples in Dixon et al. 2012 (two hESC replicates and two IMR90 replicates), with the numbers of partially filtered read pairs ranging from 60 million to 271 million. The number of partially filtered reads pairs for our samples ranged from 65 million to 236 million (sum of column of “Raw_cis #” and “Raw_trans #”), i.e. they are comparable with Dixon et al. 2012. Each sample in our cohort of cell lines has a similar amount of data, and we have a larger number of cell lines that are all of the same cell type (LCLs), which considerably increases the statistical power of our analysis.

For Hi-C data, there are then additional, more methods-specific filtering steps (e.g. removing the self-ligation fragments, removing the read pairs whose sum of distances from their mapped positions to the nearest restriction sites is larger than the length of the fragments in the Hi-C library) that have to be carried out before contact maps can be constructed. The columns “Filtered_cis #” and “Filtered_trans #” in our Supplementary Table 6 showed the number of read pairs after these additional filtering steps. This Hi-C specific filtering can lead to the removal of a large proportion of read pairs. This is the reason for the much smaller number of read pairs in these two columns – however, again, these numbers cannot be compared to those in Supplemental Table 1 from Dixon et al. 2012, and, importantly, Dixon et al. did not report what their numbers were after these Hi-C specific filtering steps.

Again, in Dixon et al. 2012 the authors did not show the number of *cis*- and *trans*-contacts after the additional, Hi-C specific steps of filtering (e.g. removing the self-ligation fragments, filtering for distance to the nearest restriction sites). But in Jin et al. 2013 (“A high-resolution map of the three-dimensional chromatin interactome in human cells”, Nature, 2013, PMID: 24141950), another landmark Hi-C study also from the Ren lab, the authors listed both the number of contacts after removing non-uniquely mapped reads and duplicates for each replicate (their Supplementary Table 1, please see below) and the number of contacts for the duplicates combined after further Hi-C specific filtering (their Supplementary Table 2, please see below). Therefore we also carried out a detailed comparison between the data from the Jin et al. paper and our Hi-C data.

Table S1. Summary of paired-end reads count of the Hi-C experiments, see **Supplementary Methods**.

	Total reads	Uniquely mapped pairs	Non-redundant pairs	Intra-chromosome pairs	Samestrand pairs	After read-level filtering		
						Samestrand pairs	Inward pairs	Outward pairs
IMR90 Rep1	397,194,480	257,464,146	185,505,290	142,888,033	36,488,898	27,934,953	19,950,119	13,944,208
IMR90 Rep2	440,242,230	275,155,361	221,966,824	127,375,768	44,780,407	33,912,506	18,519,464	17,053,882
IMR90 Rep3	621,089,009	392,648,087	225,360,969	174,286,738	16,191,349	13,691,731	11,466,490	7,053,423
IMR90 Rep4	529,157,703	355,763,158	163,466,864	130,934,899	13,206,870	11,053,352	8,922,315	5,671,266
IMR90 Rep5	234,133,577	154,952,639	127,019,197	104,863,354	12,056,254	10,025,428	8,373,062	5,093,786
IMR90 Rep6	208,710,657	132,335,077	95,775,615	72,105,868	11,823,764	9,985,535	7,015,819	5,054,531
IMR90+TNF- α Rep1	796,182,964	469,381,194	202,542,797	125,355,575	36,813,190	31,107,006	19,554,599	16,789,014
IMR90+TNF- α Rep2	607,445,170	381,403,983	304,548,370	180,400,967	72,209,998	59,601,569	32,617,090	30,033,536
IMR90+TNF- α Rep3	472,113,365	297,550,433	126,467,235	85,291,543	19,613,563	16,604,113	9,678,569	8,373,783
IMR90+TNF- α Rep4	621,694,807	395,911,149	159,726,476	108,800,646	25,949,966	22,339,959	13,483,841	11,251,514
IMR90+TNF- α Rep5	164,110,739	104,518,964	66,798,106	45,289,246	7,773,800	6,536,019	4,515,787	3,331,168
IMR90+TNF- α Rep6	229,718,795	142,486,775	107,980,643	75,776,194	17,109,744	14,729,607	9,184,622	7,405,995
IMR90 + flavopiridol	279,573,023	173,877,711	149,674,888	94,318,741	37,724,213	31,025,665	16,703,203	15,516,145
H1 (hESC)	496,522,946	335,065,429	249,998,910	212,300,490	59,755,763	45,547,792	28,526,262	22,377,390

Table S2. Summary of intra-chromosome paired-end reads count after fragment-level data filtering, see **Supplementary Methods**.

	Samestrand pairs	Inward pairs (> 1kb)	Outward pairs (> 25kb)	Total pairs
IMR90	106,603,505	52,528,106	49,671,249	208,802,860
IMR90+TNF- α	150,918,273	73,922,567	70,136,125	294,976,965
IMR90 + flavopiridol	31,025,665	15,325,775	14,288,926	60,640,366
H1 (hESC)	45,547,792	22,246,598	19,284,380	87,078,770

In Table S1 from Jin et al. 2013 (above), the percentage of partially filtered number of *cis*-contacts (i.e. after removing multi-mapping read pairs and duplicate reads) out of the raw total sequencing read pairs (Intra-chromosome pairs/Total reads in their Table S1) ranged from 15.8% to 44.8% with an average of 29.7%. The percentage in our Hi-C data is from 13.0% to 45.6% with an average of 27.1% (calculated by “Raw *cis* #” in our Supplementary Table 6 divided by “Hi-C Reads Pair Number” in our Supplementary Table 1). This shows that the percentages of *cis*-contacts out of the raw total read pairs in our data were almost the same as in Jin et al. 2013, which is strong evidence of high data quality and successful experimental work during our study.

After applying the Hi-C specific additional filtering steps, the number of *cis*-contacts decreased dramatically in Jin et al. 2013 as well. In Table S2 of Jin et al. 2013 (above), the authors only showed the number of read pairs for the replicates combined instead of for each replicate. Take IMR90 as an example. The percentage of *cis*-contact reads pairs after further filtering out of the raw total read pairs was only 8.6% (calculated as follows: $208,802,860 / [397,194,480 + 440,242,230 + 621,089,009 + 529,157,703 + 234,133,577 + 208,710,657]$) compared with 32.8% (average of Intra-chromosomal pairs/Total reads in their Table S1 for the 6 IMR90 replicates) after only partially filtering the reads. As the average percentage of *cis*-contact read pairs, after further Hi-C specific filtering, of the raw total read pairs was 8.6% for the 6 replicates in Jin et al., the percentage for individual replicates could well be lower. This demonstrates the major difference further, Hi-C specific, filtering of reads can make regarding the number of *cis*-contacts used for downstream analysis. The average percentage of *cis*-contact read pairs after further filtering out of the raw total read pairs in our Hi-C data of LCLs was 4.8%, which was slightly lower, but still comparable and in the same order of magnitude, than what Jin et al. reported for IMR90. The minor difference could be caused by factors such as cell type

specific differences (i.e. fetal lung fibroblasts in Jing et al. vs lymphoblastoid cells in our study), variance across individual cell lines, minor differences in the experimental protocols and the exact settings for the data filtering steps.

Regarding the ratio of *cis*-contacts to *trans*-contacts, the average for the partially filtered data in Table S1 of Jin et al. 2013 was 2.79 (calculated as follows: Intra-chromosome pairs/[Non-redundant pairs - Intra-chromosome pairs]) while the average for our Hi-C data was 1.12. The reason for this difference was that the percentage of partially filtered (i.e. after removing multi-mapping read pairs and duplicates) *trans*-contacts out of the raw total of sequencing read pairs was higher in our Hi-C data (27.6% compared with 12.1% in Jin et al. 2013). But this higher percentage of *trans*-contacts in our data did not come at the expense of fewer *cis*-contacts (the average percentage of partially filtered *cis*-contacts out of the raw total sequencing read pairs was 27.1% in our data vs. 29.7% in Jin et al. 2013, as mentioned above). The higher percentage of *trans*-contacts could be explained by the higher percentage of partially filtered total contacts in our data (54.6% calculated by sum of “Raw_cis #” and “Raw_trans #” in our Supplementary Table 6 divided by “Hi-C Reads Pair Number” in our Supplementary Table 1 vs. 41.7% calculated by Non-redundant pairs/Total Reads in Table S1 of Jin et al. 2013), which could in turn be the result of factors such as differences between cell types studied, and in any case remains well within the same order of magnitude as in Jin et al..

In other words, after removing multi-mapping read pairs as well as duplicates, the percentage of read pairs retained in our Hi-C data was higher than in Jin et al. 2013, indicating that our Hi-C libraries had even higher complexity and overall better quality than what was published in Jin et al. 2013, one of the landmark papers of the Hi-C field (and a study that allows for a direct comparison of parameters of Hi-C data quality relative to our data). The increased percentage of partially filtered contacts in our data stems mainly from the *trans*-contacts.

Taken together, our Hi-C data showed a very similar percentage of *cis*-contacts out of the raw read pairs when compared with Jin et al. 2013, demonstrating the high quality of the data that is available for determining *cis*-contacts. The higher percentage of *trans*-contacts is due to a higher percentage of read pairs retained after removing non-uniquely mapped reads and duplicates. This higher proportion of *trans*-contacts enabled us to investigate the global changes in chromosome contacts, genome-wide, in the 22q11.2 deletion cells.

We would also like to clarify that in the revised manuscript we did not use the global Hi-C data to carry out the topological domain calling and A/B compartments analysis as this global Hi-C data can only show the conformation changes for the combination of two homologous chromosomes (which again is the inherent nature of Hi-C data) while the Reviewers encouraged a haplotype-specific analysis of features such as topological domains. Therefore our analyses of topological domains and A/B compartments were based on the newly generated, very deep, capture Hi-C data in combination with the haplotype phasing data.

Regarding the Reviewer’s comment that “*ID00016 map is missing data over several Mb in the 22q13.1-2 region*”, we would like to clarify that the haplotype-specific contact maps shown in Figure 2 were constructed based on the newly generated capture Hi-C data rather than the Hi-C data. In the haplotype-specific contact map only regions with sufficiently high numbers of heterozygous SNVs can be phased with the combined linked-read/familial genotypes based approach we were using. This requirement is leaving a gap in the phasing in the region from 39

Mbp to 42 Mbp in ID00016. In other words, the absence of signal for region is not caused by low coverage of the underlying Hi-C data (our capture Hi-C data was equivalent to 7-12 billion genome-wide Hi-C sequencing reads for each cell line, i.e. it is of very deep coverage [please also see the original reply to the same Reviewer below]) but rather the Hi-C data for this region could not be phased, as indicated in line 240-241 in the legend of Figure 2. However, while this region is on chromosome 22q it is far distant from the 22q11.2 deletion region and was not a region of interest in our study, also when looking across our entire cohort. The lack of phasing in this region does not affect our integrative analyses or findings in general, and again is not an indicator of quality issues with our Hi-C data.

The new experimental work for generating the very deep capture Hi-C data for two patient and two control cell lines was performed with *in-situ* Hi-C protocol reported in Rao et al. (Cell 2014) [PMID: 25497547]. Therefore, to further assess the quality of the data in our study, we also compared our capture Hi-C data with the data from Rao et al., by using a number of quality control metrics as reported in Rao et al. 2014. In the table below are the quality control metrics of our capture Hi-C data (metrics from Rao et al. 2014 are also listed, in the second column; ‘Intra restriction fragment’ denotes the percentage of read pairs where both ends align to the same fragment; ‘Ligations’ denotes the percentage of read pairs containing the ligation junction sequence GATCGATC of MboI; ‘Read Pair type’ denotes the percentage of intra-chromosomal contacts by read pair type: L — both reads map to the reverse strand, R — both reads map to the forward strand, I — two reads map to different strands and point towards each other, O — reads map to different strands but point away from one another):

	Rao et al. 2014	ID00014	ID00016	GM12878	GM12892
Uniquely mapped Reads	80%-91%	81%	79%	78%	81%
Duplicates	1%-13%	2%	2%	3%	2%
Intra restriction fragment	1%-6%	3%	3%	3%	2%
Inter chromosomal	21%-31%	53%	60%	57%	55%
Intra chromosomal	69%-79%	47%	40%	43%	45%
Intra Short Range (<20 kb)	27%-37%	15%	12%	16%	18%
Intra Long Range (≥20 kb)	63%-73%	85%	88%	84%	82%
Ligations	25%-47%	32%	32%	31%	32%
Read Pair type	25%-25%-	25%-25%-	25%-25%-	25%-25%-	25%-25%-
(L-I-O-R)	25%-25%	25%-25%	25%-25%	25%-25%	25%-25%

All the quality control metrics of our capture Hi-C data aligned well with the corresponding numbers for the Hi-C libraries of Rao et al. 2014 except that the percentage of inter-chromosomal contacts was higher and that of intra-chromosomal contacts was lower. In fact, the QC metric of percentage of long-range intra-chromosomal contacts, which is a crucial quality metric of Hi-C experiments, was even better in our data than in Rao et al. 2014. However, the higher percentage of inter-chromosomal contacts and the lower percentage of intra-chromosomal contacts were to be expected as only the contacts involving chromosome 22 were captured and sequenced. Therefore a larger proportion of inter-chromosomal contacts out of the genome-wide inter-chromosomal contacts was captured than were intra-chromosomal contacts out of the genome-wide intra-chromosomal contacts.

To be explicit, suppose the length of chromosome 1,2,...,22 is denoted by x^1, x^2, \dots, x^{22} . Then the proportion of intra-chromosomal contacts on chromosome 22 out of the genome-wide intra-

chromosomal contacts can be approximated as $x^{22}/(x^1+x^2+\dots+x^{22})$ which we round up to 2%. This means only 2% of the intra-chromosomal contacts in the Hi-C libraries would be kept after capture. However, the proportion of inter-chromosomal contacts involving chromosome 22 out of the genome-wide inter-chromosomal contacts can be roughly calculated as $(x^1+x^2+\dots+x^{21})/((x^1+x^2+\dots+x^{21}) + (x^1+x^2+\dots+x^{20}) + (x^1+x^2+\dots+x^{19}) + \dots + (x^1+x^2) + x^1)$
 $= (\sum_{i=1}^{21} x^i) / (\sum_{i=1}^{21} (22 - i)x^i)$. Now we only need to verify if $(\sum_{i=1}^{21} x^i) / (\sum_{i=1}^{21} (22 - i)x^i) > 2\%$. As $(\sum_{i=1}^{21} x^i) / (\sum_{i=1}^{21} 50 * x^i)$ is equal to 2% and $(\sum_{i=1}^{21} 50 * x^i)$ is much larger than $(\sum_{i=1}^{21} (22 - i)x^i)$, it becomes clear that $(\sum_{i=1}^{21} x^i) / (\sum_{i=1}^{21} (22 - i)x^i)$ is $> 2\%$.

This means that a smaller proportion of the intra-chromosomal contacts out of the genome-wide intra-chromosomal contacts in the Hi-C libraries were captured in the capture Hi-C data than were inter-chromosomal contacts. This explains the difference of intra- and inter-chromosomal contacts between global Hi-C data and our capture Hi-C data. Taken together, our capture Hi-C data, which we used for the haplotype-specific construction of contact maps and the analysis of A/B compartments and topological domains, were of high quality.

5. Also the more detailed and informative breakdown into short and long-range *cis*-contacts is missing from Supplemental Tab. 6, which is informative, as the Hi-C signal is mainly generated by the $>20\text{kb}$ intra-chromosomal contacts.
 Supplemental Table 1 from Dixon et al. 2012 (see attachment)

The breakdown of short and long-range *cis*-contacts of our Hi-C data is as below.

Sample	# Cis -contacts after removing multi-mapping reads and duplicates	Intra-chromosomal $> 20\text{kb}$	Intra-chromosomal $< 20\text{kb}$
GM06990	33,008,728	5,460,767	27,547,961
GM07939	83,987,223	33,279,151	50,708,072
GM10847	52,580,552	6,527,623	46,052,929
GM12878	81,706,427	25,810,998	55,895,429
GM12892	81,379,308	34,795,506	46,583,802
GM17938	68,291,638	13,006,130	55,285,508
GM17942	69,229,887	12,399,814	56,830,073
GM18505	27,868,756	5,225,752	22,643,004
ID00014	47,343,395	8,574,326	38,769,069
ID00015	44,238,248	7,493,759	36,744,489
ID00016	160,874,295	11,608,838	149,265,457

The average percentage of long-range *cis*-contacts ($>20\text{kb}$) out of the total number of the *cis*-contacts for our Hi-C data was 21.9%. In Dixon et al. 2012, this percentage was 19.3%, 42.7%, 38.8% and 55.3% for the four human cell lines respectively (hESC Original, hESC Replicate, IMR90 Original and IMR90 Replicate). The percentage for our data was relatively low but still within the range. Also there is again the consideration of cell type specific differences for this metric (i.e. we used LCLs for our study while Dixon et al. 2012 used hESCs and the IMR90 cell line).

For additional confirmation that our Hi-C data QC metrics are within the normal range we downloaded from GEO (accession number GSE43070) all the files of the mapped non-redundant read pairs for Hi-C data from Jin et al. 2013 and counted the number of short and long range *cis*-contacts. Here is the breakdown of all the samples:

Sample	Intra-chromosomal < 20kb	Intra-chromosomal > 20kb	Percentage of intra-chromosomal > 20kb
GSM1154021_HiC.IMR90.rep3	287,401,698	61,171,778	17.5%
GSM1154022_HiC.IMR90.rep4	212,405,116	49,464,682	18.9%
GSM1154023_HiC.IMR90.rep5	164,108,068	45,618,640	21.8%
GSM1154024_HiC.IMR90.rep6	99,317,364	44,894,372	31.1%
GSM1154027_HiC.IMR90_TNF.rep5	61,135,642	29,442,850	32.5%
GSM1154028_HiC.IMR90_TNF.rep6	86,276,484	65,275,904	43.1%
GSM1154025_HiC.IMR90_TNF.rep3	96,146,990	74,436,096	43.6%
GSM1055800_HiC.IMR90.rep1	157,288,022	128,488,044	45.0%
GSM1154026_HiC.IMR90_TNF.rep4	118,565,758	99,035,534	45.5%
GSM1055805_HiC.H1.rep2	216,920,806	207,680,174	48.9%
GSM1055802_HiC.IMR90_TNF.rep1	118,493,822	132,217,328	52.7%
GSM1055801_HiC.IMR90.rep2	87,174,522	167,577,014	65.8%
GSM1055803_HiC.IMR90_TNF.rep2	70,346,296	208,882,588	74.8%
GSM1055804_HiC.IMR90_Flav.rep1	46,560,280	142,386,074	75.4%

While the percentage of the long-range *cis*-contacts out of the total number of *cis*-contacts could be as high as 74.8%, it could also be as low as 17.5%. There is a large degree of variance between experimental conditions (e.g. with or without TNF) and between cell types (i.e. again IMR90 cells and hESCs). And even among replicates for the same cell line and experimental condition the variance is very striking (e.g. for the 6 replicates of IMR90, the percentage could be as high as 65% and as low as 17.5%). The value for our data for this metric, 21.9%, falls within the range reported in this landmark paper as well.

Taken together, the percentage of the long-range *cis*-contacts of our Hi-C data was entirely within the normal range. Differences in exact values between our Hi-C data and the published data may be caused by factors such as the differences for this metric in different cell types, minor differences in the way the experimental procedures were carried out or the exact settings on the computational pipelines.

Of note, we used the capture Hi-C data (and not the global Hi-C data) for the haplotype-specific analysis of interactions on chromosome 22, of the A/B compartments and of the topological domains. Our capture Hi-C data had a very high percentage of long-range *cis*-contacts (82%-88%), as mentioned above.

6. The sub-optimal quality of the primary Hi-C data does not devalue all analyses of the manuscript at hand, but produces a very low resolution Hi-C experiment and very much limits the possible analyses to physical separation at A/B compartment level and general interaction frequencies across the genome.

As described in the sections above, the comparative analysis that was suggested by the Reviewer, of our Hi-C data and the data from several landmark papers from the Hi-C field, shows that the

quality of our data is on par with the Hi-C data from these landmark papers. Therefore there are no reasons to expect the analyses in our study to be devalued or to be only possible at very low resolution compared to the state-of-the-art in the field. Our analyses and results are according to the current standards of the field (and exceed those standards with the haplotype-specific portions of the study). We again thank the Reviewer for suggesting this comparison to the data from the landmark Hi-C papers, which we think increases the confidence in our results even further.

This view is further supported by the fact that a specific prediction of a change in chromosome folding behavior that resulted from the analysis of our Hi-C data, i.e. the change in interaction between the downstream deletion proximal region and the telomere proximal region on 22q, could be validated by an entirely independent and orthogonal method, 3D FISH.

We would like to clarify again that the haplotype-specific analysis of interactions on chromosome 22, of the A/B compartments and of the topological domains were based on the capture Hi-C data (which was also of high quality as described above) rather than the global Hi-C data. Regardless, as mentioned above, the quality and the resolution of our global Hi-C data was also comparable with the previous publications such as Jin et al. 2013 especially for the percentage of *cis*-contacts of the total number of read pairs. The earlier landmark publications such as Jin et al. 2013 did not use their Hi-C data for haplotype-specific analysis (it would not have been of deep enough coverage for this purpose and furthermore it is not clear if there were phased genome sequences available for the cell lines used in those earlier studies). To achieve the very deep coverage with high quality Hi-C data that is necessary for phased Hi-C analyses we added high-depth capture Hi-C data to our study. Regarding the ability to predict changes in general interaction frequencies across the genome, please see our answer to Reviewer comment 8., below.

7. In the revision process the authors nicely show that the reported switch in A/B compartments from the original version of the manuscript was due to a problem with the data analysis.

The A/B compartment change reported in the original manuscript was due to the inherent inability of global Hi-C data at standard depth of coverage and mapped to a non-phased genome sequence to distinguish the Hi-C signal coming from the two homologous chromosomes. However, this non-phased Hi-C analysis is indeed currently state-of-the-art of the field. In the revision our phased Hi-C analysis then overcame this limitation of what is the current standard in the field. As such the finding regarding the switch in A/B compartments from the original version of the manuscript was due to a limitation in the data analysis, but not because we did not carry out the original analysis properly, rather it was specifically because we carried out the initial analysis according to the standards of the field. When we generated additional deep-coverage capture Hi-C data and phased the cell line genome sequences, after prompting by the Reviewers, we were able to overcome this limitation in the current state-of-the-art.

As we have also mentioned in our response to Reviewer's comment #2 above, the A/B compartment analysis was based on the newly generated capture Hi-C data in a haplotype-specific way rather than the re-evaluation of the global Hi-C data.

Overall this outcome is a very encouraging example for how the peer-review process leads to improved results and better standards for cutting edge fields of research, and we want to thank the Reviewer again for encouraging us to undertake the phased Hi-C analysis.

8. Also, I am wary to believe the global changes in trans-contacts (Fig. 6 and S12), although tested with permutation tests, until replicated with higher-quality Hi-C data.

We would like to restate that, as shown by the comparison to landmark Hi-C papers described above, and by the independent validation of a Hi-C-predicted interaction change using 3D FISH, our Hi-C data is of a quality that compares well with current Hi-C standards, e.g. as reported in Jin et al. 2013, and therefore we submit that there is no reason to reject the analyses shown in Figures 6 and S12 based on general concerns about data quality. In fact, the percentage of *trans*-contacts of the total number of read pairs was higher in our Hi-C data than in that of Jin et al. 2013, due to better mapping rates and lower numbers of duplicates.

This enabled us to obtain numbers of *cis*-contacts that are comparable to those in Jin et al. 2013, but more *trans*-contacts even at similar sequencing depth. This higher number of *trans*-contacts empowered our analysis of the global changes in *trans*-contacts without compromising the *cis*-contacts analysis. Therefore, since the analysis of global changes is based on high quality Hi-C data and the findings are supported by permutation tests, we feel that this is a result that can be reported to the field, where it should stimulate discussion and an examination of whether such changes in the global Hi-C patterns also occur in other contexts, e.g. in cells with other large CNVs, and what the molecular mechanisms are that are mediating such changes.

9. The authors state that the TADs adjacent to the microdeletion do not change. As mentioned above, I find this very hard to extract from the Hi-C data generated in this study, however, I do believe that the authors are correct with their statement. The 22q11 deletion region is located in between several clusters of repeats/microsatellites that are a) difficult to map onto a reference human genome and b) likely the breakpoints for the recurrent deletions at this locus. Consequently, even in control individuals these microsatellites are possibly some sort of boundaries of flanking new topological and/or regulatory domains and the flanking domains should be not affected. High-quality Hi-C experiments in combination with the already generated ChIP-seq data would help to study this in detail.

We are encouraged that the Reviewer concurs that our findings show that the TADs adjacent to the microdeletion do not change. We agree that there is a limitation to the extent to which functional genomics signals such as Hi-C or ChIP-Seq coming directly from the immediate sequence neighborhood of the 22q11.2 deletion breakpoints can be resolved, because of the stretches of segmental duplication sequence in this area. This is indeed a function of the nature of the reference genome in these directly breakpoint-surrounding neighborhoods coupled with the current short-read (e.g. Illumina type) DNA sequencing technology.

As we have detailed in the above the quality of our Hi-C data is on par with that from landmark Hi-C studies. However, as pointed out by the Reviewer above, mapping Illumina sequencing reads from any kind of experimental assay onto the segmental duplication sequences in the human reference is subject to limitations, and this is true also for Hi-C data even if it is of the highest quality. We chose to err on the side of caution and map the reads with stringent criteria to avoid the cross-mapping that can otherwise occur on segmental duplications. Therefore we do not expect the Hi-C signal to be resolving up to the exact breakpoints of the large CNV (e.g. at nucleotide resolution for that given stretch of sequence).

However, because of the nature of our analyses of the changes of chromatin conformation in 22q11DS, which aim to detect large-scale and long-range effects, this feature of the deletion on

22q11.2 being directly bounded by segmental duplication sequences has no detrimental impact on our study. There are extended stretches of unique genome sequence, of lengths matching the lengths of genomic sequence along which TADs extend (i.e. often hundreds of kbp in length), that are right adjacent to these segmental duplication sequences. As we have shown in this study we can map our Hi-C and ChIP-Seq data onto these unique sequences with confidence and then determine large-scale features of chromosome conformation such as TADs or long-range interaction changes, and the effects the large CNV has on these features.

As a perhaps interesting side note, even the emerging long-read sequencing technologies, such as the Pacific Biosystems platform (which in any case would not be amenable for a study such as ours here because of the low number of sequencing reads produced by these platforms), have only very limited potential to resolve genomic segmental duplications sequences [Chaisson et al., Nature 2015, PMID: 25383537].

We would like to clarify again that the haplotype-specific TADs in the revised manuscript were calculated using the capture Hi-C data rather than the Hi-C data. Both the capture Hi-C data and the Hi-C data were of high quality as mentioned above. Based on the haplotype-specific TADs using the capture Hi-C data, we observed that the interactions between the TADs adjacent to the deletion were increased (Supplementary Figure 6a-b). However, these increased interactions were not strong enough to form a newly fused TAD (Supplementary Figure 6a-b). We concur with the Reviewer that the exact boundaries of the TADs flanking the deletion are most likely situated in the segmental duplication regions bounding the 22q11.2 deletion (Supplementary Figure 6).

10. I would refrain from an epigenetic analysis for differentially enriched ChIP-seq signals as performed in Figure 3a-c. As the authors point out correctly in their reply to my earlier comments, individual regions/enhancers/promoters can behave differently from one another and make the analysis “inconclusive”. Plotting the \log_{10} -pvalue of \log_2 -transformed fold-changes in “[500kb] bins with significantly differentially enriched sites against the background of the whole genome.” (lines 888-895) will produce a result that will not reflect what happens in the deletion flanking regions on an epigenetic level. All the reader can take from such an analysis is that there is some sort of change in the regions flanking the breakpoints, but essentially the reader is left alone with the interpretation with no possibility to evaluate the data themselves.

The Reviewer’s comment regarding our Figure 3a-c, that “*there is some sort of change in the regions flanking the breakpoints*”, sums up in a general sense what we meant to report in the manuscript with regard to these particular ChIP-Seq signals. As we discuss in line 261-273 in the version of the revised manuscript with changes tracked, we observed consistent global changes of chromatin modifications in the flanking regions of the deletion (all but one of the differentially bound sites of H3K27ac were bound less and all the differentially bound sites of H3K27me3 were bound more). This indicates that across the individual regulatory elements in these deletion-flanking regions there is consistent change in their histone marks, and these systematic epigenetic changes were conclusive and statistically significant across the deletion-flanking regions.

We also report that these epigenetic changes for entire, extended deletion flanking regions were not completely or consistently reflected by the gene expression changes in the LCLs. But

this does not weaken the observation of changes across entire and extended deletion flanking regions on the epigenetic level.

Furthermore we submit that this finding of epigenetic marks changing across entire and extended regions outside of a large deletion CNV is not just interesting in its own right (and is pointing towards follow-up studies e.g. in other cell types, and for other large CNVs), but also it allows for the intriguing speculation that there may be interaction between the histone marks and chromosome conformation changes that we report for these same extended regions. Specifically, we report that the 500kbp windows across which the histone marks change – are also the windows for which the Hi-C signals change. Therefore the finding is that across an entire 500kbp window outside of the large deletion CNV both histone marks and chromosome folding (and TAD-interactions) are changing, which we submit is interesting, novel, important to report, it should lead to hypotheses about the molecular mechanisms that may be at play and it adds to the knowledge in the chromosome folding field in general.

With regards to the reader being able to evaluate the data themselves, all the data will be available for download from the NCBI Gene Expression Omnibus (GEO; <http://www.ncbi.nlm.nih.gov/geo/>) under accession number GSE76922.

We again thank the Reviewers for the extensive comments and suggestions during both rounds of review, which have substantially improved this manuscript.

Reviewer #3 (Remarks to the Author):

I would like to thank the Authors for their extensive reply to my comments. I was hesitant to write the review as directly as I did, because implying bad craftsmanship (as in: "I'm not sure whether the Hi-C worked as good as it should have.") is a serious comment and I did not do this lightly. The better it is when answered by an extensive analysis as was done in this case.

After sieving through these analyses, I agree with the authors that the Hi-C experiments are on par with published data and as such the conclusions drawn from all further analysis in this manuscript are in fact strengthened.

Two of my other comments were unrelated to this observation, namely the changed inter-chromosomal interaction patterns and the presentation of the ChIP-seq data.

I stand by my comments regarding both observations. I'm still not convinced about the specific changes in inter-chromosomal interactions. Nevertheless, I find the question addressed and discussed adequately.

Also, the data analysis and presentation of the ChIP-seq is not helpful for any reader. However, regarding the ChIP-seq analysis, I would suggest editing the results section in order to lead the reader through the analysis (Just as a note, it took me several minutes to decipher how you have processed the ChIP-seq data).

Maybe something like this ll. 246ff:

"To do so we performed enrichment analysis in our ChIP-seq data do detect significantly differential signals of H3K27ac, H3K27me3, and CTCF enrichment in 500kb bins. We found that the deletion-flanking regions were significantly enriched for differences in H3K27me3 and H3K27ac signal as compared to control LCLs. For CTCF, only the distal deletion-flanking region was differentially enriched.

We then looked more specifically, which individual regions within the differential 500kb bins contributed to the differential signal. Within the proximal" l.261

Although the authors have a short sentence in the discussion, I think it is necessary to point out the rather inconclusive nature of the observations in the results.

Finally, I must add an minor but important objection with regards to data presentation. The overall Figure design and layout should be improved in order to communicate the findings quickly and easily comprehensible to the interested readership. Although I read Hi-C and genomics papers more or less on a daily basis, some of the figures took a bit to decipher, only for their unusual data presentation.

This concerns, for example, the plotting of genomic data for chr 22 in Fig. 1e-h and Fig. 3 a-c - a pictogram of the chromosome like in other figures would signal immediately what is plotted here. Also having the two magnified areas in Fig. 1a matching each other would help. A maybe more important point is the plotting of the Hi-C data in Fig. 1, S5, and Fig 4 (to a lesser degree). Aside from the fact that this is the first time I have seen this color scheme whilst presenting Hi-C data, with red-green blindness affecting ~5% of the population these colors are not helpful - a red-blue scale as in other figures would be more appropriate.

Last but not least I must add that I was very happy with the form of the peer review that happened on this manuscript because, although critical and direct in content, I had the feeling that the authors understood my concerns tried to address them constructively and not in a confrontative manner. I would appreciate if the review process ends up published alongside the manuscript (not least because the 15 pages reviewer response full with data analysis would end up published and not only buried somewhere in our email accounts).

In the below please find our point-to-point response to the reviewer's comments. The revisions not mentioned in the below were made per editorial requests.

REVIEWERS' COMMENTS:

Reviewer #3 (Remarks to the Author):

1. I would like to thank the Authors for their extensive reply to my comments. I was hesitant to write the review as directly as I did, because implying bad craftsmanship (as in: "I'm not sure whether the Hi-C worked as good as it should have.") is a serious comment and I did not do this lightly. The better it is when answered by an extensive analysis as was done in this case. After sieving through these analyses, I agree with the authors that the Hi-C experiments are on par with published data and as such the conclusions drawn from all further analysis in this manuscript are in fact strengthened.

We very much appreciate the Reviewer's comments. This was one of the most thorough review processes that most of the authors have experienced, and we feel it was a rather rewarding experience. On our part we would like to thank the Reviewer again for the review process in general, which lead to a markedly improved manuscript, and specifically for prompting us to perform the extensive analyses of the quality control metrics of our Hi-C and capture Hi-C data. We think this was a very useful exercise indeed, regarding the confidence in the results reported in the manuscript, and also given the available option of having the entire correspondence made publicly available – others may find this discussion about proper QC metrics of Hi-C experiments useful and it might contribute to further solidify QC standards in the field.

2. Two of my other comments were unrelated to this observation, namely the changed inter-chromosomal interaction patterns and the presentation of the ChIP-seq data. I stand by my comments regarding both observations. I'm still not convinced about the specific changes in inter-chromosomal interactions. Nevertheless, I find the question addressed and discussed adequately.

We thank the Reviewer for this comment. We had decided to include the findings on the global changes in chromosome interactions as they were based on high quality Hi-C data, data which produced intra-chromosomal findings that could be validated by FISH, and as the findings were supported by permutation tests. We reasoned that this is a result we learned from the current data and it should stimulate discussion and further investigation in the field. However, we did indeed take care to elaborate how the results were obtained and discuss the findings very carefully with the goal of avoiding overreaching and we are very happy to find that the Reviewer judges that this was done in an adequate fashion.

3. Also, the data analysis and presentation of the ChIP-seq is not helpful for any reader. However, regarding the ChIP-seq analysis, I would suggest editing the results section in order to lead the reader through the analysis (Just as a note, it took me several minutes to decipher how

you have processed the CHIP-seq data). Maybe something like this II. 246ff: "To do so we performed enrichment analysis in our CHIP-seq data to detect significantly differential signals of H3K27ac, H3K27me3, and CTCF enrichment in 500kb bins. We found that the deletion-flanking regions were significantly enriched for differences in H3K27me3 and H3K27ac signal as compared to control LCLs. For CTCF, only the distal deletion-flanking region was differentially enriched. We then looked more specifically, which individual regions within the differential 500kb bins contributed to the differential signal. Within the proximal" I.261

We have found this point very helpful. We have revised the corresponding sections as suggested.

4. Although the authors have a short sentence in the discussion, I think it is necessary to point out the rather inconclusive nature of the observations in the results.

We modify the original sentence and added one more sentence to make it clear that the CTCF binding signal only changed in the distal flanking region. The revised part now reads as follows: Interestingly, both distal and proximal flanking regions of the 22q11.2 deletion were enriched with differential signals for the histone marks H3K27ac and H3K27me3 while only the distal flanking regions were enriched with differentially binding sites of CTCF. At the present the reason for this discrepancy is not known. We note that only the deletion-distal flanking region is engaging in increased intra-chromosomal interactions with the telomeric end of chromosome 22q but the available data is not sufficient to conclude that differential binding of CTCF is causally involved in this phenomenon.

5. Finally, I must add a minor but important objection with regards to data presentation. The overall Figure design and layout should be improved in order to communicate the findings quickly and easily comprehensible to the interested readership. Although I read Hi-C and genomics papers more or less on a daily basis, some of the figures took a bit to decipher, only for their unusual data presentation. This concerns, for example, the plotting of genomic data for chr 22 in Fig. 1e-h and Fig. 3 a-c - a pictogram of the chromosome like in other figures would signal immediately what is plotted here. Also having the two magnified areas in Fig. 1a matching each other would help.

We thank the Reviewer for these specific comments. We have added ideograms of chromosome 22 in Fig. 1e-h, Fig. 3a-c and Fig. 7b, and have matched the two magnified heatmaps in Fig. 1a. We also have changed the design and layout for Fig. 1a, 1c and Fig. 8.

6. A maybe more important point is the plotting of the Hi-C data in Fig. 1, S5, and Fig 4 (to a lesser degree). Aside from the fact that this is the first time I have seen this color scheme whilst presenting Hi-C data, with red-green blindness affecting ~5% of the population these colors are not helpful - a red-blue scale as in other figures would be more appropriate.

This is, of course, a very important point, we were not aware of this issue before and are thankful that it has been brought to our attention. We have changed the colors in these figures

to red-blue scale (which we will now also use in all future manuscripts) as suggested by the Reviewer. We also recolored Fig.3, Fig.5, Fig.7 and Fig.8.

7. Last but not least I must add that I was very happy with the form of the peer review that happened on this manuscript because, although critical and direct in content, I had the feeling that the authors understood my concerns tried to address them constructively and not in a confrontative manner. I would appreciate if the review process ends up published alongside the manuscript (not least because the 15 pages reviewer response full with data analysis would end up published and not only buried somewhere in our email accounts).

We can only echo this statement in its entirety. We again thank the Reviewer for the extensive effects put into our manuscript during the three rounds of review, which have substantially improved this manuscript. We took the Reviewer's concerns very seriously and tried our best to address by conducting further analyses. The exchanges were direct and critical in content but always constructive. We very much agree with the reviewer that the comments of the Reviewers and our response letters should be published along with the manuscript. It could be useful for the readers.

We hope that the above responses have addressed the comments and suggestions made by the Reviewer. Like the Reviewer mentioned above, and again, we are also very glad to see how the whole peer review process has considerably strengthened our manuscript. We thank the Reviewer again for the constructive critiques and comments.